# Multi-indicator sustainability assessment of global food systems

Abhishek Chaudhary [1], David Gustafson [2] & Alexander Mathys [1]

Food systems are at the heart of at least 12 of the 17 Sustainable Development Goals (SDGs). The wide scope of the SDGs call for holistic approaches that integrate previously "siloed" food sustainability assessments. Here we present a first global-scale analysis quantifying the status of national food system performance of 156 countries, employing 25 sustainability indicators across 7 domains as follows: nutrition, environment, food affordability and availability, sociocultural well-being, resilience, food safety, and waste. The results show that different countries have widely varying patterns of performance with unique priorities for improvement. High-income nations score well on most indicators, but poorly on environmental, food waste, and health-sensitive nutrient-intake indicators. Transitioning from animal foods toward plant-based foods would improve indicator scores for most countries. Our nation-specific quantitative results can help policy-makers to set improvement targets on specific areas and adopt new practices, while keeping track of the other aspects of sustainability.

[1] Sustainable Food Processing Laboratory, Institute of Food, Nutrition and Health, ETH Zurich, Schmelzbergstrasse 9, Zurich 8092, Switzerland. [2] Real Whirlwind, LLC, 63141, Saint Louis, Missouri, USA. Correspondence and requests for materials should be addressed to A.C. (email: abhishek.chaudhary@hest.ethz.ch)

Meeting the increasing demand for nutritious food in the face of growing world population, consumption levels, dietary shifts, and the consequent environmental degradation constitute a major challenge for humanity in this century. Food systems are by far the biggest employer in the world, particularly in economically poor countries. They also represent an important part of national gross domestic product (GDP), although the share of income generated from agriculture production, food processing, sales, and distribution varies across different countries. Going forward, global food systems need to ensure improved economic security of actors involved and combat existing malnutrition/obesity-related health problems, while keeping the environmental impacts low enough so as not to transgress the planetary boundaries of biophysical processes and further destabilize Earth systems[1]. Most of the United Nations' 2030 Sustainable Development Goals (SDGs) are linked with global food systems performance[2]. Although global in scope, the SDGs require concerted efforts at the national and regional level, and, consequently, each nation needs to build concrete food policy pathways and implement measures customized to its local conditions[3].

A logical first step in building national transformative pathways is to select appropriate indicators that can evaluate and track the progress toward the sustainability of its diets and food systems.

However, a comprehensive global sustainability assessment of national food systems taking into account multiple domains of interest such as nutrition, environment, economic, social, and resilience is currently lacking[4]. Studies estimating environmental impact associated with dietary intake rarely consider their nutritional values and are often limited to either a particular country[5,6] or a single environmental indicator[7–9].

A number of global studies have suggested that dietary changes such as reducing the consumption of animal-sourced food and adopting vegetarian options can lead to reduced environmental impact in the form of reduced greenhouse gas (GHG) emissions, water use, land use, and reduced risk of diet-related non-communicable diseases[10–16]. However, these global studies are also limited to one or just a few environmental indicators (mostly GHG emissions) and typically consider only a specific nutritional aspect of diet such as total caloric or protein intake. Thus, these previous studies have not considered micronutrients, whose deficiency (so called "hidden hunger") affects over two billion people worldwide[17]. In sum, studies that are global in scale and evaluate food systems using multiple indicators of sustainability are rare[10,11].

Gustafson et al.[18] recently proposed a novel comprehensive framework to quantitatively characterize the performance of national food systems through seven metrics of sustainable nutrition security as follows: (1) Food Nutrient Adequacy; (2) Ecosystem Stability; (3) Food Affordability and Availability; (4) Sociocultural Wellbeing; (5) Food Safety; (6) Resilience; and (7) Waste and Loss Reduction (see Table 1). These metrics were selected through consensus building activities involving a number

### Table 1 Seven food system metrics, their indicators, and data sources

| Metric | Indicator | Median | Source | GDP correlation |
|---|---|---|---|---|
| Food Nutrient Adequacy | | **61** | | 0.53 |
| | Shannon Diversity of Food Supply | 74 | Remans et al.[33] | 0.42 |
| | Non-Staple Food Energy | 46 | Remans et al.[33] | 0.72 |
| | Modified Functional Attribute Diversity | 77 | Remans et al.[33] | 0.70 |
| | Population Share with Adequate Nutrients | 76 | This study | 0.64 |
| | Nutrient Balance Score | 75 | This study | 0.46 |
| | Disqualifying Nutrient Score | 12 | This study | − 0.74 |
| Ecosystem Stability | | **47** | | − 0.36 |
| | Ecosystem Status | 43 | Hsu et al.[34] | 0.51 |
| | Per-Capita GHG Emissions | 51 | This study | − 0.79 |
| | Per-Capita blue water consumption | 50 | This study | − 0.75 |
| | Per-Capita Land Use | 50 | Alexander et al.[9] | − 0.09 |
| | Per-Capita Non-Renewable Energy Use | 28 | World Bank[59] | 0.00 |
| | Per-Capita Biodiversity Footprint | 50 | Chaudhary et al.[28] | 0.02 |
| Affordability and Availability | | **63** | | 0.83 |
| | Food Affordability | 54 | GFSI[37] | 0.85 |
| | GFSI Food Availability Score | 56 | GFSI[37] | 0.80 |
| | Poverty Index | 88 | GFSI[37] | 0.82 |
| | Income Equality | 62 | World Bank[60] | 0.24 |
| Sociocultural Wellbeing | | **60** | | 0.71 |
| | Gender Equity | 68 | WEF[39] | 0.43 |
| | Extent of Child Labor | 50 | ILO[40] | 0.59 |
| | Respect for Community Rights | 60 | WRI[41] | 0.63 |
| | Animal Health and Welfare | 60 | API[42] | 0.70 |
| Resilience | | **57** | | 0.64 |
| | ND-GAIN Country Index | 52 | Chen et al.[43] | 0.80 |
| | Food Production Diversity | 64 | Remans et al.[33] | −0.20 |
| Food Safety | | **71** | | 0.76 |
| | Global Burden of Foodborne Illnesses | 50 | WHO[45] | 0.70 |
| | Food Safety Score | 88 | GFSI[37] | 0.80 |
| Waste and Loss Reduction | Pre- and Post-Consumer Food Waste and Loss | **68** | FAO[46] | −0.68 |

The global median value of each indicator (normalized to 0–100 scale) across 156 countries is shown. Metric score in bold is the arithmetic average of its underlying indicator scores (see Supplementary Data 1 for all indicator values per country). Spearman's rank correlation (ρ) between GDP per capita and indicator values of different countries is also shown. See Supplementary Data 5 for correlation value of all 25 × 25 indicator combinations

of nutrition, economic, food system, and climate change experts representing a range of global public and private institutions (see Gustafson et al.[18] and Acharya et al.[19] for details).

Each of the metrics comprises multiple indicators that are combined to derive an overall score (0–100). However, the initial application of the metrics was limited to nine countries. Here we address this research gap and quantify the food system sustainability status of 156 countries for the year 2011 using the 7 metrics and 23 underlying indicators proposed by Gustafson et al.[18] along with two additional indicators on biodiversity impacts and health-sensitive nutrient intake.

For the Food Nutrient Adequacy metric, we for the first time report the Nutrient Balance Score (NBS)[20] of each country by comparing national daily average intake amounts of 25 essential (qualifying) food nutrients with their Reference Daily Intake values (see Methods).

Based on the Disqualifying Index (DI) proposed by Fern et al.[20], we also calculate a new indicator, the Disqualifying Nutrient Score (DNS), by comparing the total daily intake of four public health-sensitive food nutrients (sugar, cholesterol, saturated fat, and total fat) with their Maximal Reference Values

(MRVs). Countries with lower intake of these sensitive nutrients score higher on this indicator.

Next, we estimate each country's Population share with Adequate Nutrients (PANs) by comparing the per capita daily food nutrient supply to a demographically weighted threshold for a population through EAR-"cut-point" approach[21]. The national daily average intake of each nutrient was obtained by combining a food composition database[22] with the FAO food balance sheet (FBS) database[23] and adjusted for non-edible and wasted food[24].

For the Ecosystem Stability metric, we calculate the per capita carbon[25] and water footprint[26,27] of each nation's daily diets (year 2011). We also assembled the land and biodiversity footprint of each country's daily food consumption based on calculations by Alexander et al.[9], and Chaudhary and Kastner[28], respectively.

For the rest of the five metrics (Food Affordability and Availability, Sociocultural Wellbeing, Food Safety, Resilience, and Food Waste), we collected the data for each indicator from a number of sources and converted them to 0–100 scale[18].

In order to explore the sustainability outcomes and potential trade-offs between different indicators, we progressively excluded

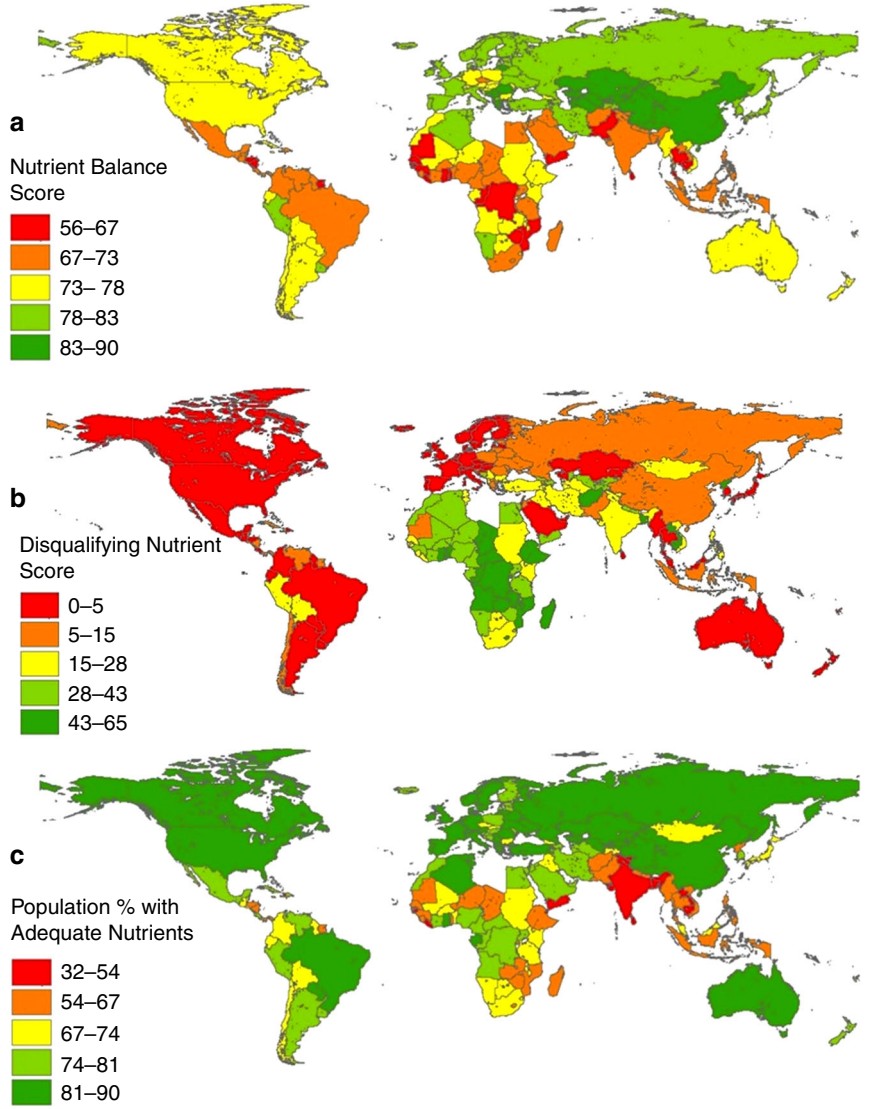

**Fig. 1** Food nutrient adequacy indicator scores for 156 countries. **a** Nutrient Balance Score, **b** Disqualifying Nutrient Score, and **c** Population Share with Adequate Nutrients for all countries calculated in this study (year 2011). Countries for which data were not available do not appear on the map (e.g., Somalia). Symbology based on Jenks natural breaks classification method. See Supplementary Data 1 for all values per country

animal-sourced foods from current diets and constructed three alternative dietary scenarios [healthy global diets (HGD), lacto-ovo vegetarian (VGT), and vegan (VGN)][11] for each of the 156 countries based on global dietary guidelines on healthy eating[5,11,29–32]. For these alternative diets, we recalculated three nutritional (NBS, DNS, and PAN) and two environmental (carbon and water footprint) indicator scores for each country.

The food system metrics and individual indicator scores presented here make it possible to holistically measure the current status of national food system performance, identify key areas of improvement, and provide valuable information for country-specific policies aimed at designing sustainable transformation pathways. Our analysis reveals that indicator scores vary widely among different countries depending upon their geographical location, income status, and dietary habits. Transitioning to more plant-based diets would primarily improve both nutrition indicators, and carbon and water footprints, but might need to be accompanied by micronutrient supplementation.

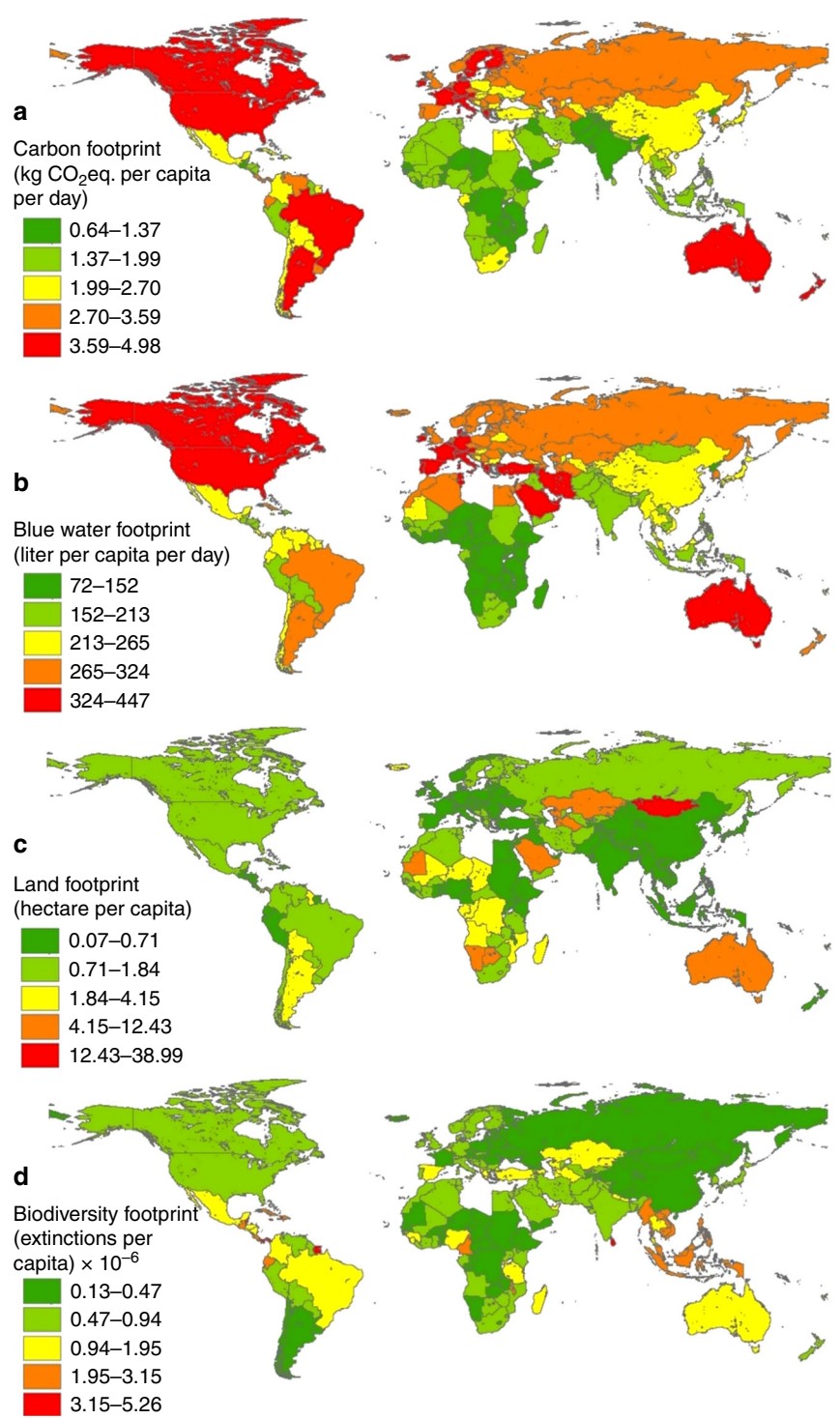

**Fig. 2** Per capita food related environmental footprints of 156 countries. **a** Carbon, **b** Blue water, **c** Land, and **d** Biodiversity footprint of daily average diets of each country in the year 2011. Symbology based on Jenks natural breaks classification method. See Supplementary Data 1 for normalized (0–100) score of indicators

## Results

**Food Nutrient Adequacy.** We found that the global median Food Nutrient Adequacy score for the year 2011 was 61 (Table 1) with minimum and maximum values of 46 (Cambodia and Bangladesh) and 69 (Costa Rica and Greece). Regarding the individual indicators, the world median Shannon Diversity of Food Supply (reflecting how many different types of food items are consumed in a country)[33] and Modified Functional Attribute Diversity (MFAD, reflecting the diversity in nutrients provided by the different food items)[33] is ~ 75% (see Supplementary Data 1 for all scores). However, the Non-Staple Food Energy (indicating the proportion of energy derived from food items other than cereals, roots, or tubers)[33] was quite low at 46%. Shannon Diversity of Food Supply showed relatively small variation across 156 countries, MFAD showed moderate, whereas Non-Staple Food Energy showed high geographical variation (~ 33% for Sub-Saharan Africa and ~ 66% in North America; Supplementary Data 2).

For the fourth indicator, PAN, we found that the global median value was 75.73% (averaged over 17 essential nutrients) (Table 1). In terms of individual nutrients, the % adequacy is low globally for Vitamin D, Vitamin E, Calcium, and folate (Supplementary Data 3). The calculated world median value for the NBS[20] came out to be 75. The geographic pattern for PAN and NBS is very similar to that of MFAD with higher scores for EU and North America, and lower values for Sub-Saharan and south Asian population (Fig. 1a and Supplementary Data 1). However, we found that the trend in DNS was almost the opposite with higher values for Sub-Saharan Africa, Middle-East, and South Asia. The DNS was lowest for Europe, Latin America, and North America (Fig. 1), as their daily intake of four nutrients of concern often exceeds corresponding MRVs (see Supplementary Data 2).

Interestingly, we found a statistically significant correlation between the GDP per capita (hereafter denoted simply as "GDP") of nations and the first five nutrient adequacy indicators (Spearman's rank correlation, $\rho > 0.4$), suggesting the positive effect of higher income on national nutritional diversity and adequacy (Table 1). However, there was a strong negative correlation between DNS and GDP ($\rho = -0.74$, Table 1). We found that all individual indicators are positively correlated with each other, except DNS, which is negatively correlated with the other five indicators (see Supplementary Data 5 for Spearman's correlation coefficient between all indicators).

**Ecosystem Stability.** As shown in Table 1, this metric is defined as the average of six indicators—the Environmental Performance Index[34], which quantifies the status of ecosystems within a country's border, together with five other indicators based on the per capita diet related environmental footprint. We found that mean dietary carbon footprints vary from around 0.7 kg $CO_2$ eq. per capita per day for African countries such as Mozambique, Ethiopia, and Malawi to > 4 kg $CO_2$ eq. per capita per day for New Zealand, Australia, USA, France, Austria, Argentina, and Brazil (Fig. 2a and see Supplementary Data 1 for uncertainty range). Consequently, the normalized score on the scale of 0–100 for the former countries is > 80 compared with < 30 for the latter (Supplementary Data 1).

We also compared the calculated per capita food related GHG emissions with their "sustainable levels," as recently defined by Roos et al.[35] They assumed that food consumption could use no more than 50% of the yearly total per capita emissions of ~ 1000–2000 kg $CO_2$ eq., which have been defined as sustainable by the Intergovernmental Panel on Climate Change (IPCC). This target corresponds to ~ 750 kg $CO_2$ eq. per capita per year (2.055 kg $CO_2$ eq. per capita per day). Currently, 81 of 156 countries (representing ~ 52% of global population) have mean dietary

carbon footprints above this threshold (Fig. 2a). However, this comparison is not statistically significant due to high uncertainty in the emission factors of food products[25]. The calculated national carbon footprints vary over an order of magnitude around this threshold (Supplementary Data 1).

Bovine meat consumption contributes the most to the diet related GHG emissions for most regions, except Asia, where rice contributes to > 30% of daily per capita GHG emissions (see Supplementary Data 4). Globally, other major contributors to GHG emissions include milk, poultry, pig meat, goat meat, and beer. For Asia, consumption of coconut and fish products, and for Sub-Saharan Africa, consumption of bovine meat, yams, plantains, and fermented beverages, were identified as being major contributors to GHG emissions in addition to rice (Supplementary Data 4).

The pattern for Per-Capita diet related blue water footprint is similar to that of dietary carbon footprints with Sub-Saharan Africa averaging around 140 l per capita per day compared with ~ 350 for North America and 300 l per capita per day for Europe (Fig. 2b). However, although the average carbon footprint of Middle-East and North Africa was 35% low2868er than Europe, the food related blue water footprint of these two regions is similar (~ 300 l per capita per day). The major food items contributing the most to diet related freshwater consumption globally are rice, wheat, and sugar. Region-specific food items contributing toward high blue water footprint include the following: olive oil and pig meat in Europe, dates and nuts in Middle East and North Africa, bovine meat in North America, spices in South Asia, tea in Latin America, and fruits and maize in East Asia (Supplementary Data 4).

Per-Capita diet-related Land Use or "land footprint" is highest for countries sourcing animal food produced from extensive ruminant grazing such as Mongolia (39 ha per person), Namibia (12), Kazakhstan (11), Mauritania (11), and Australia (8) compared with countries such as Bangladesh, India, Pakistan, Sri Lanka, Indonesia, and Philippines that have high population and relatively less animal-sourced foods in their diets (< 0.3 ha per person; Fig. 2c).

On average, 66% of the total land footprint of African and Latin American countries comprises pasture land, compared with just 33% for South Asian and 50% for East Asian and European countries (see Table S1 for the % of total land footprint due to pasture and crop land per country).

Regarding the Per-Capita Biodiversity Footprint of national food consumption, tropical Central American and Caribbean countries such as Belize, Suriname, Panama, Cuba, Jamaica, and Haiti, all have high per capita impacts because of relatively high species richness per unit area, small populations, and low import levels[28,36], which results in higher species loss as the natural habitat is converted to agriculture land use for food production (Fig. 2d). Western European and North American countries also have high impacts because of their high consumption levels and imports of food items from tropical countries "embodying" high species loss[28]. Fig. 2 shows that nations with higher land footprints do not necessarily have higher biodiversity footprints (e.g., Mongolia), because certain land uses such as extensive livestock grazing are less harmful to species than intensive cropland[28,36]. Juxtaposing national Food Nutrient Adequacy and Ecosystem Stability metric scores (Fig. 3) reveals that most nations with high nutritional quality also have high environmental footprints.

**Food Affordability and Availability.** Figure 4 shows the metric score for different world regions aggregated based on geographical location and income levels (see Supplementary Data 1

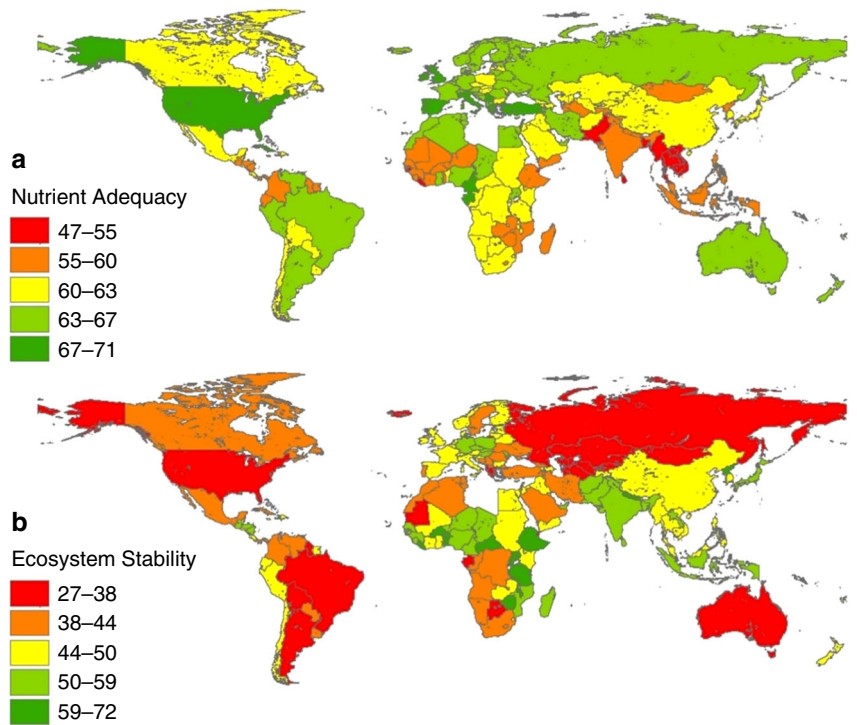

**Fig. 3** Nutrition and Environmental metrics scores per country. **a** Food Nutrition Adequacy and **b** Ecosystem Stability scores for all countries calculated in this study (based on data for the year 2011). Metric score is arithmetic average of its underlying indicator scores (normalized to 0–100). Symbology based on Jenks natural breaks classification method. It can be seen that most countries with high Food Nutrient Adequacy score low on Ecosystem Stability, pointing toward high environmental impacts associated with their diets (see Supplementary Data 1 for all values)

for results per country). We found that this metric shows a high positive correlation with national GDP levels ($\rho = 0.83$, Table 1), ranging from > 80 for high-income nations such as the EU, UK, USA, Canada, Australia, and Japan to around 30 for low-income Congo, Haiti, Malawi, and Madagascar. Latin American and South Asian countries stood in the middle at 63 and 50, respectively (Fig. 4). The first two indicators of this metric—Food Affordability (percentage of household expenditures on items other than food)[37] and Food Availability (ease of physical access to food)[37]—reflect factors such as economic status and disposable income that usually increase with a country's economic growth and increase in the capacity of consumers to purchase nutritious food. As expected, the third indicator, Poverty Index[37], defined as the proportion of each country's population living above the poverty line ($1.90 per day), has lower values in low-income countries. Interestingly, the last indicator Income Inequality (as measured by the Gini Coefficient)[38], which has a value of 100 for the case of perfect income equality and 0 for the case of all income earned by a single individual, is not as strongly correlated with GDP ($\rho = 0.24$, Table 1).

**Sociocultural Wellbeing.** Sociocultural Wellbeing metric ranges from ~ 90 for high-income Western Europe nations such as Iceland, UK, Norway, and Finland to < 40 for Chad, Benin, Mali, and Burkina Faso (Fig. 4 and Supplementary Data 1). Although the previous two metrics dealt with quantifying environmental impacts and economic access for consumers, this metric reflects the status of societal factors (the third so-called "pillar" of sustainability). All four indicators, Gender Equity[39], Extent of Child Labor[40], Respect for Community Rights[41], and Animal Health and Welfare[42], were positively correlated with each other as well as with GDP (Supplementary Data 2). The one showing the weakest correlation with GDP is Gender Equity ($\rho = 0.43$, Table 1).

**Resilience.** The first indicator within the resilience metric, Notre Dame Global Adaptation Initiative (ND-GAIN) Country Index, reflects a country's vulnerability (exposure and sensitivity to the negative effects of climate change) and readiness (ability to leverage investments and convert them to adaptation actions), and is calculated based on the status of 45 different factors (e.g., flood hazard)[43,44]. It increased with increasing GDP (average score of 70 for high-income nations compared with 36 for low-income countries). The other indicator—Shannon Diversity of Food Production[33]—reflects the expected contribution to a country's resilience by having more than just a few crops being produced. It actually showed a small negative correlation with income levels and is more uniformly distributed around the world (Fig. 4). The overall effect of these two complementary indicators is that the resilience score of high-income nations was just higher (~ 10 points) than the rest (Fig. 4 and Supplementary Data 1).

**Food Safety.** The Food Safety metric showed very high values (~ 90) for high-income nations compared with tha of low-income nations (~ 20) (Fig. 4 and Supplementary Data 1). The first indicator within the Food Safety metric, the Global Burden of Foodborne Illnesses, provides mortality, morbidity, and disability-adjusted life years associated with each major food-borne disease[45], and has lower values in Sub-Saharan Africa and South Asian countries owing to tropical climate, lower income, and the relative lack of medical facilities and resources to deal with these illnesses. The second indicator, Food Safety Score[37], has higher values for countries where a food safety regulatory agency and a formal grocery sector is present, and where a higher percentage of the population has access to potable water, all of which are positively correlated with income levels.

**Waste and Loss Reduction.** This metric quantifies the portion of the produced food that is not either lost (pre-consumer) or

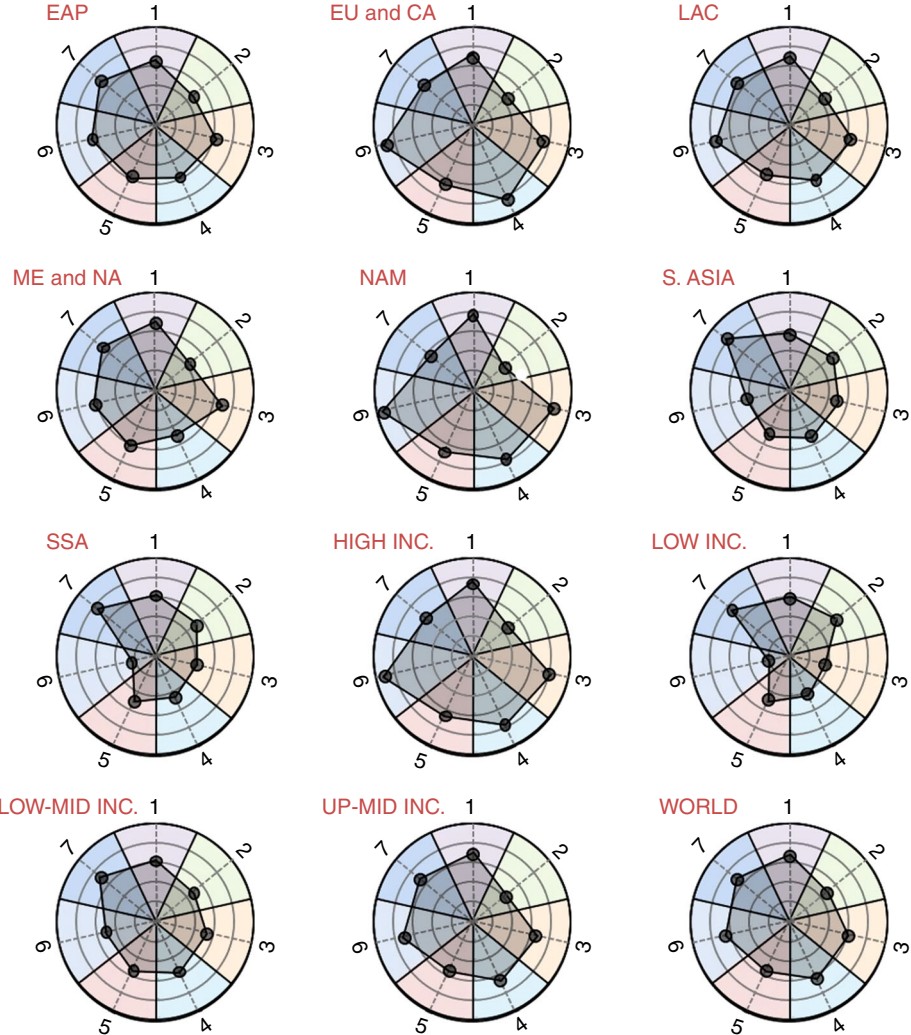

**Fig. 4** Average food system metric scores for different World Bank regions and income levels. The areas of the polygons represent relative national status of food system sustainability (higher the better). The concentric circles are at intervals of 20. 1. Food Nutrient Adequacy; 2. Ecosystem Stability; 3. Food Affordability and Availability; 4. Sociocultural-Wellbeing; 5. Resilience; 6. Food Safety; 7. Waste and Loss Reduction. Regions are as follows: East Asia and Pacific (EAP); Europe and Central Asia (EU&CA); Latin America and Caribbean (LAC); Middle East and North Africa (ME&NA); North America (NAM); South Asia (S. Asia); Sub-Saharan Africa (SSA), and world. Income levels: high, low, lower-middle (LOW-MID INC.), and upper-middle (UP-MID INC)

wasted (post-consumer) in a country[46]. High-income countries such as Canada, USA, Australia, and EU countries score lower (~ 60) than low- or lower-middle income countries such as Mongolia, India, Sri Lanka, Indonesia, etc. with scores > 80 (Fig. 2 and Supplementary Data 1).

**Alternative dietary scenario outcomes**. Adoption of alternative diets HGDs, VGT, and VGN leads to high improvement in DNS across Europe, Latin America, and North America (Table 2). This positive change is a consequence of major reduction in animal sourced food from their current levels, which in turn implies lesser intake of disqualifying nutrients (total fat, saturated fat, and cholesterol)[20].

The PAN score shows widely varying changes across the world: increasing for Asia and Sub-Saharan Africa, decreasing for Europe and North America, while remaining almost unchanged for Latin America and Middle East (Table 2). This is primarily due to an increase in caloric intake for Asia and Sub-Saharan Africa (regions with extensive current undernourishment) and a decrease in caloric intake in Europe and North America (regions with currently higher than recommended caloric intake) under

the three scenarios (see Methods). The HGD scenario improves the % adequacy of almost all 17 nutrients in most countries (except for some high income nations), whereas the VGT scenario shows slight improvement in % adequacy of most nutrients except Vitamin $B_{12}$. Adoption of VGN diets considerably improves the % adequacy of folate, magnesium, and Vitamin C due to higher fruit and vegetable intake (see Supplementary Data 3) but also leads to a high increase in deficiency risk for Vitamin $B_{12}$ and Selenium for all countries. This is because these micronutrients are primarily supplied by meat, fish, and dairy products in most countries, whose intake is reduced to zero under VGN scenario (Supplementary Data 3).

The patterns of changes in NBS mirror that of PAN but are relatively smaller in magnitude (Table 2). This is because if the daily requirement for a specific qualifying nutrient is already being met, any further increase in intake of that nutrient under the alternative scenario does not serve an additional nutrition function and therefore does not increase the NBS (see Eq. 1 and 2 in Methods)[20]. However, the NBS decreases if the intake amount of a specific qualifying nutrient decreases below its Reference Daily Intake levels, such as in the case of Vitamin $B_{12}$ and Selenium under VGN (Supplementary Data 3).

**Table 2 Region-specific changes in nutritional indicator values**

| Region | Scenario | NBS | DNS | PAN | Carbon footprint | Water footprint |
|---|---|---|---|---|---|---|
| East Asia and Pacific (EAP) | REF | 74 | 14 | 68 | 2.33 | 225 |
| | HGD | +3 | +4 | +9 | +0.02 | +30 |
| | VGT | +1 | +10 | +5 | −0.58 | +27 |
| | VGN | −1 | +34 | +2 | −0.69 | +38 |
| Europe and Central Asia (EU&CA) | REF | 80 | 10 | 82 | 3.04 | 303 |
| | HGD | +1 | +3 | −4 | −0.89 | −68 |
| | VGT | 0 | +10 | −5 | −1.48 | −79 |
| | VGN | −4 | +50 | −12 | −2.01 | −85 |
| Latin America and Caribbean (LAC) | REF | 73 | 7 | 72 | 2.36 | 229 |
| | HGD | +5 | +5 | +5 | −0.17 | −10 |
| | VGT | +4 | +15 | +2 | −0.88 | −19 |
| | VGN | 0 | +43 | −3 | −1.14 | −18 |
| Middle East and North Africa (ME&NA) | REF | 75 | 19 | 77 | 1.97 | 302 |
| | HGD | +2 | +4 | −1 | −0.32 | −42 |
| | VGT | +1 | +13 | −5 | −0.88 | −42 |
| | VGN | −1 | +33 | −8 | −1.10 | −36 |
| North America (NAM) | REF | 77 | 0 | 84 | 4.25 | 348 |
| | HGD | +3 | 0 | −9 | −1.74 | −121 |
| | VGT | +2 | +10 | −9 | −2.64 | −134 |
| | VGN | −1 | +50 | −16 | −3.12 | −138 |
| South Asia (S. Asia) | REF | 68 | 24 | 56 | 1.52 | 207 |
| | HGD | +6 | +4 | +17 | +0.45 | +69 |
| | VGT | +5 | +6 | +15 | +0.03 | +74 |
| | VGN | +3 | +23 | +12 | −0.16 | +81 |
| Sub-Saharan Africa (SSA) | REF | 71 | 38 | 68 | 1.48 | 141 |
| | HGD | +3 | −1 | +9 | +0.24 | +35 |
| | VGT | +1 | +3 | +5 | −0.25 | +33 |
| | VGN | 0 | +13 | +3 | −0.37 | +34 |

DNS, Disqualifying Nutrient Score; HGT, healthy global diets; NBS, Nutrient Balance Score; PAN Population Share with Adequate Nutrients; VGN, vegan; VGT, lacto-ovo vegetarian. The increase (+) or decrease (−) in indicator values under three alternative dietary scenarios, HGD, VGT, and VGN, are presented. Changes in dietary carbon (kg $CO_2$ eq. per capita per day) and water footprints (liters per capita per day) are also shown. *REF corresponds to the indicator values under current diets. See Supplementary Data 6 for changes in values per country *NBS, DNS, and PAN are scaled indicators (0–100) of the Food Nutrient Adequacy metric (see Table 1). The region-aggregated indicator scores were calculated by taking the average of all national indicator scores within the region. To put the values into context, a NBS and DNS value of 100 implies perfectly nutritious diets. A PAN score of 100 implies that 100% of the region's population is meeting daily nutritional requirements (see Methods for details). The current global average carbon and water footprint is 2.3 kg $CO_2$ eq. per capita per day and 237 l per capita per day, respectively

scenarios are very similar. For example, the current blue water footprint (reference scenario) of North America is 348 l per capita per day, whereas under the HGD, VGT, and VGN scenarios it is 227, 214, and 210 l per capitaper day, respectively (i.e., a reduction of 35%, 38%, and 40%, respectively). This is because most of the difference is accounted for in the green water (rain water) component (not considered here). The differences between blue water requirements for animal- and plant-based food items are much smaller than the differences in their carbon footprints (e.g., carbon footprint of bovine meat is 50 times higher than that of wheat but its blue water footprint is just two times higher)[26,27]. We found that the geographic trend in water footprint largely parallels changes in carbon footprints, with an increase of 20–40% for Asia and Sub-Saharan Africa and a reduction of 20–40% for Europe and North America under all three scenarios.

## Discussion

It is essential that food system metrics employed to measure the current or future sustainability status are intrinsically holistic and capable of detecting trade-offs across different economic, environmental, and social elements. We present a first global-scale analysis that quantifies national food system sustainability status through the use of 25 indicators across 7 metrics (Figs 1–4). The results show that different countries have widely varying patterns of performance with unique priorities for improvement.

Apart from the overall metric scores (Fig. 4), the individual results on sustainability indicators (Supplementary Data 1) provide useful insights to all countries on many fronts. For example, the average intake of four health-sensitive nutrients: sugar, saturated fat, total fat, and cholesterol is currently 2, 2, 2, and 7 times higher for high-income countries than the low-income countries[23]. The % population adequacy of Vitamin D, Vitamin E, Calcium, and Folate is low even in high income countries (Supplementary Data 3). Such information is valuable in designing different interventions, such as increased production of particular crops[47], dietary diversification[33], food imports[48], biofortification[49], tax incentives[50], etc. Again, such interventions to improve nutrition must consider the consequences for other sustainability elements (e.g., impacts on environment, rise in food prices, and unemployment).

Our results on current levels of per capita diet-related carbon, water, land, and biodiversity footprints (Fig. 2) can be a starting point for individual countries to design improvement strategies to reduce food consumption impacts on the environment. For each of the 156 countries, we present results for all food products that contribute >10% to their total carbon and water footprints (Supplementary Data 4), which can help nations consider alternative foods that would lower their dietary footprints. Although not explicitly included by Gustafson et al.[18], we included a separate indicator for the national per capita biodiversity footprint because of the following reasons: biodiversity is one of the 9 elements in the planetary boundary framework[1], the current rate of species extinction is over 100 times the background rate[51], and anthropogenic land used for agriculture is one of the major drivers of global biodiversity loss, and that the land footprint is not a good proxy for the biodiversity footprint[28].

The analysis on nutrition–environmental outcomes of dietary changes provides several interesting insights (Table 2). First, the results indicate that dietary changes toward fewer animal and more plant-based foods can result in significant reductions in daily per capita disqualifying nutrient intake (e.g., >50 point increase in DNS for North America and Europe) but relatively small improvements in the NBS. Second, we found that adopting diets low in animal-sourced foods can significantly reduce the food-related per capita GHG emissions and freshwater use of

Regarding the environmental consequences, adoption of HGD, VGT, and VGN diets would result in global average food-related per capita GHG emissions that are 12, 40, and 50% lower than under current diets, respectively.

Dietary changes in North America and Europe lead to the largest reductions in per capita carbon footprint primarily because of a decrease in total caloric intake and reduced meat intake levels under the three scenarios (Table 2). The carbon footprint increases slightly for South Asia and Sub-Saharan Africa under the HGD scenario (by 0.45 and 0.25 kg $CO_2$ eq. per capita) because of an increase in calories from current levels.

We found that compared with the current (reference) scenario, the blue water footprint decreased under the HGD, VGT, and VGN scenarios (except for countries whose caloric intake had to be scaled up to 2,300 kcal levels). However, unlike GHG emissions that decreased substantially from HGD to VGN, the national water footprints under the three HGD, VGT, and VGN

countries (e.g., > 70% and 40% reductions, respectively, under the VGN scenario for North America and Europe).

Third, despite above promising signs, we also find that the above dietary change scenarios do not always lead to twin nutrition and environmental benefits for all countries through all indicators. For example, although adoption of HGD in South Asia has the potential to increase PAN score by an additional 17%, it also entails an increase in per capita diet-related GHG emissions and water use by 0.45 kg $CO_2$ eq. per capita per day and 69 l per capita per day, respectively (Table 2). Such results suggest that diets higher in nutrition are not necessarily more environmentally beneficial. Similarly, a reduction in the GHG and water footprint for North America and Europe under these scenarios comes at a cost of decrease in PAN (Table 2). This is because these regions currently have high intake of meat products (> 100 g per capita per day), which is scaled down to 42 g per capita per day under the HGD scenario and down to 0 g under VGN and VGT scenarios (see Methods). The scaling up of the intakes of fruit, vegetables, or pulse crop consumed currently is insufficient to compensate for some of the micronutrients (Vitamin $B_{12}$, Vitamin E, Selenium, and Calcium) provided mainly by meat products consumed under the current diets (Supplementary Data 3). This suggests that dietary guidelines on less meat (e.g., VGT and VGN) should potentially be accompanied by additional recommendations on the intake of special plant-based foods rich in particular micronutrients such as Brazil nuts for Selenium[52] or dried purple laver for Vitamin $B_{12}$[53].

On the socio-economic results, we acknowledge an important limitation in the underlying indicators upon which we rely[18], namely, the lack of any measures of the overall economic health of the various factors within the food system itself, i.e., producers, food transport and processing, retail, food service industry, etc.— all being highly relevant. Having said this, our results for the economic and social components of sustainability are largely consistent with expectations. In general, the Food Affordability and Availability increases with national income level, with a concomitant reduction in poverty. However, income inequality is not as strongly correlated with GDP (Table 1), meaning that rising national income levels do not uniformly decrease income disparities within countries. The Sociocultural Wellbeing metric and all of its underlying indicators are positively correlated with national incomes (Table 1 and Supplementary Data 5), but the weakest correlation is with gender equity, suggesting that women's empowerment concerns persist even as national incomes rise. As pointed out elsewhere[44], the resilience and performance of food systems can be efficiently enhanced by focusing interventions on vulnerable populations, which would be reflected in improvements of the gender equity and income inequality indicators. Taken together, these results suggest that policies encouraging stronger national incomes will improve the economic and social aspects of food system sustainability.

The same is true for Food Safety, which is also very strongly correlated with GDP. However, neither the Resilience nor the Food Waste metrics show such a strong trend (Table 1). Although higher-income countries tend to have less vulnerable and more resilient infrastructures, they also tend to focus their food production on a very small number of crops[33], potentially making them more susceptible to extreme weather or other forms of "system shocks." Efforts to diversify food production and reduce post-consumer food waste are most urgently needed in higher-income countries, whereas reducing pre-consumer food loss and improving food safety need the greatest attention in lower income countries[24].

The study is first, to our knowledge, to quantify the nutritional quality of global diets considering multiple micro- and macro-nutrients and several indicators (Fig. 1). Overall, our results and

trends in the GHG and water-use consequences of dietary change are consistent with previous studies, who also found that reducing animal-sourced food leads to reduction in national water[8,14,54] and carbon footprints[10,11,15,55]. Differences in absolute values of the footprint between ours and previous studies might occur due to a number of factors such as different year of the analysis, use of different dietary scenarios, different food product emission factors[25], inclusion or not of land-use change in product carbon emission factors, etc.

The input data used to calculate seven food system metrics and different indicators come from several sources, with uncertainties and limitations that should be considered when interpreting the results. Using FAO's national FBS[23] as a source for per capita daily food and nutrient intakes within a country is not ideal, because the actual number of food items consumed in a country can be quite large and the FBS provides information on only 94 food items, many of which are aggregated and can vary considerably in nutrient composition depending upon their regional varieties and cooking style. It was not possible to characterize this uncertainty, but in order to minimize error in this step, we followed the standard guidelines[56,57] to match FBS products with products in the food composition database[22] (see methods). Next, we used nutrient values of different food items from a single food composition database[22] for all countries, although the nutrient composition of similar food items may vary geographically. As detailed national food intake data and localized food composition tables become available, such uncertainties can be reduced.

On the environmental side, one limitation of this study is that we used global average blue water footprint values for different food items[26,27]. Using country-specific footprint values for individual food items (where available) will improve the accuracy of the results. We found that the carbon footprint values per country have high uncertainty, varying by over an order of magnitude (Supplementary Data 1). For all other indicators, it was not possible to quantify the uncertainty owing to lack of underlying data.

For calculating the Resilience metric, we employed the Shannon diversity of food production[33] and the ND-GAIN index[43]. The ND-GAIN index is composed of a vulnerability score and a readiness score, and represents the best available overall indicator of resilience. However, additional factors might be reasonable to include, especially ones relating to preparedness to drought and coastal flooding[18] which can be important in many places and should be explored by future studies.

Our list of 25 indicators is not exhaustive, but a key criterion in the selection of these indicators is that they can be derived from data that are either directly available for all countries (e.g., food affordability and availability)[37] or can be readily estimated by processing global databases such as FBS[23] (e.g., NBS, see Methods above). This means that the indicator and metric scores can be calculated and regularly updated for subsequent years to track the progress of national food system sustainability.

We acknowledge that alternative indicators (e.g., human development index or International Food Policy Research Institute's (IFPRI) Global Hunger Index) could be relevant substitutes for some employed by us here, but it was beyond the scope of this study to defend specific indicators or replace them, as this would have again entailed consensus building activities involving experts from diverse backgrounds. Instead, we simply utilized a complete set of indicators that had been fully described and justified in very recent studies[18–20,28] as being fully relevant for characterizing the status of global food sustainability.

We could not account for the sub-national variation in food systems issues such as food insecurity and therefore acknowledge the limitation of our country-average indicator and metric scores in policy making. Indeed, the indicators would have much higher

practical utility if reported at sub-national scales and separately for different age, sex, and different socio-economic groups of the population. However, this would require significant data collection efforts at much finer spatial resolution than currently available.

Future studies should extend our work by quantifying the impact of food systems through additional indicators (e.g., nitrogen and phosphorus emissions causing eutrophication and acidification) and comparing them with their planetary boundaries (sustainable levels)[1]. Combining our metric-based methodology with future scenarios generated by integrated assessment models could provide insights into the impact of various climate change scenarios, socio-economic development pathways, demographic and diet trends, and alternative global trade policies on future global food systems[18].

We constructed three simple dietary change scenarios[11] with constraints on fruits, vegetables, red meat, sugar, and total caloric intake, and evaluated the sustainability outcomes through just five indicators. We could not evaluate the full implications that such dietary transformations would have on the environment, and social and economic aspects. For example, increasing the intake of vegetables/fruits and replacing animal-sourced foods with vegetarian substitutes might adversely affect the income of small livestock farmers in some regions. In certain cases, such substitutions can also increase the carbon footprint in colder countries due to an increase in energy use for producing vegetables in greenhouses[54]. Designing more sophisticated scenarios with constraints on other food groups, considering consumer preferences[58], and employing additional nutritional, environmental, economic, and social indicators of sustainability could provide new insights and represent important areas for future research.

Our nation-specific quantitative results can be used by national and global policy-makers to identify key areas of improvement and set priorities. The multi-indicator approach enables one to evaluate the impact of alternative strategies aimed at a particular aspect (e.g., reducing carbon footprint by replacing ruminant meat with dairy products or pulses), while at the same time monitoring the impact on other food system performance metrics (e.g., Food Nutrient Adequacy). The holistic analysis presented and demonstrated here thus helps advance the sustainability evaluation framework, as well as provide insights into the potential impact of food system interventions intended to improve both human and planetary health. Such food system transformation actions by individual countries and regions are key to attaining the newly adopted global SDGs[2].

## Methods
**Overview.** Data for the indicators associated with metrics 3–7 (Food Affordability and Availability, Sociocultural Wellbeing, Food Safety, Resilience, and Waste and Loss Reduction) per country were directly imported from various sources and converted to a 0–100 scale using Eq. 5 below. Please refer to Table 1 for sources of data for all indicators. It is noteworthy that some of these national indicator scores are determined not only by food systems but other sectors/industries as well (e.g., mining). For this study, we specifically considered the impacts of national food systems alone where the available data allowed us to do so (e.g., carbon, water, land footprints associated with national food consumption alone). However, a lack of data did not allow us to do so in case of six out of 25 indicators selected in this study. These exceptions include the following: non-renewable energy use[59], poverty index[37], income equality[60], gender equity[39], respect for community rights[41], and ND-GAIN index[43] (see ref. [18] for full details on each indicator). For some indicators, such as food waste and losses, the data are available for certain aggregated world regions only and needed to be downscaled to national level for this study, potentially compromising its accuracy[46].

The national values for the first three indicators of food nutrition adequacy metric (Shannon Diversity of Food Supply, MFAD, and the Non-Staple Food Energy) were taken directly from Remans et al.[33]. For all other indicators of the first two metrics (Food Nutrition Adequacy and Ecosystem Stability), we describe the methods to calculate each indicator below.

**National food consumption data.** We obtained the data on national level supply of 94 food commodities (g per capita per day) for the year 2011 from the FBS of the FAO[23]. These amounts are food reaching the consumer (on an "as purchased" basis), i.e., as the food leaves the retail shop or otherwise enters the household. The quantities are provided on the basis of "primary equivalents." For example, amount of bread is reported as wheat equivalent. We converted the food product supply values to actual consumption (intake) values by correcting for non-edible food mass (e.g., bones in meat) as well as household waste using conversion factors from Gustavsson et al.[24]

**Nutritional profile of FAO food items.** The 94 food items from FBS food supply data[23] were matched with the food commodities in the USDA SR28 food composition database[22] using standard guidelines[56,57] to estimate the amount of calories and other food nutrients they supply. The 94 food commodities consist primarily of individual food items, such as "rice" or "apples," and some aggregates of foods such as "Vegetables, other". For single food commodities, the assignment of nutrient values was straightforward. For example, the nutrient values for the FBS[23] commodity "apples" were imported directly by locating the respective item (e.g., "apples, raw") from the USDA database[22].

For each of such individual commodities, we documented the calories as well as amount (g/g) of the following 25 essential (qualifying)[20] food nutrients: Folate, Niacin, Pantothenic acid, Riboflavin, Thiamin, Calcium, Copper, Iron, Mg, Mn, P, K, Se, Zn, Linoleic acid, Choline, Dietary Fiber, Protein, and Vitamins A, B-6, B-12, C, D, E, and K. The four nutrients of concern (disqualifying nutrients)[20] included in this analysis and recognized as potentially having adverse effects on health and for which necessary data are available were as follows: sugar, total fats, saturated fats, and cholesterol. For items that are normally consumed in a cooked form, such as vegetables, the nutrient values for the cooked item were selected from the food composition tables and matched to the FBS commodity, as there are weight and nutrient losses in the food during cooking[57].

For aggregate commodities such as "Vegetables, other", FAO[23] provides a list of 427 possible individual foods (e.g., spinach, cauliflower, etc.), which make up this aggregated commodity (see Supplementary Data 7). To estimate the proportion of the total weight attributed to each single food in the aggregate commodity for each of 156 countries, we followed the guidelines recommended by Stadlmayr et al.[56] and applied in Arsenault et al.[57]. To this end, the information from FAO on production and trade amounts of 427 single commodities was used[23]. For all individual foods in an aggregate commodity, the production and import amounts were added for the country, and the export amount was subtracted to obtain the amount in the country's food supply. Next, the nutrient values of the aggregate commodities were calculated as the weighted average of the nutrient values of each individual food items within the aggregate (i.e., based on the proportion attributed to each food within the aggregate)[57].

Instead of taking simple mean or median of all single foods within an aggregated commodity, the above weighted average procedure is more suitable, because it allows for taking into account the differences between countries on their intake of individual food items (e.g., one nation might have higher spinach intake, whereas the other have high carrot intake, although both belong to the FAO's aggregated commodity "Vegetables, other")[23,57]. Nutrients such as vitamins A, C, and folate are present in varying and sometimes large amounts in the individual foods comprising the FAO's aggregated commodities such as "Fruits, other," "Vegetables, other," "Pulses, other and products," etc.[23,57].

Although dietary intake surveys provide better information on consumed food items, processing methods and intake amounts, but such data are not available for all countries. Moreover, care should be taken on their use as some studies have shown that household survey data also have some limitations[61]. The FBS data are already available and collected routinely; they offer an informative and relatively easy starting point to compare likely national diets. Future studies could employ data from new sources, such as the global dietary database (http://www.globaldietarydatabase.org/), which are currently under development and aim to provide more refined national dietary intakes of food and nutrients for children and adults by age, sex, pregnancy/nursing status, rural vs. urban residence, and education levels.

**Estimating national nutrient intake amounts.** Finally, by multiplying the calories and nutrient values per gram of all 94 food items by their intake amounts (in g) and summing up, we obtained the total daily caloric intake ($E_k$ in kcal per capita per day) and daily intake of each of the 25 qualifying nutrient $q$ ($a_{q,k}$ in g per capita per day), and each of four disqualifying nutrient $d$ ($a_{d,k}$ in g per capita per day) for each country $k$.

**Nutrient Balance Score.** We calculate the NBS ($0 < NB < 100$) of each country's daily diet as:[20]

$$NB_k = 100 \cdot \left( \frac{\sum_{q=1}^{25} QI_{q,k}}{N_q} \right) \qquad (1)$$

where $QI_{q,k}$ is the value for the Qualifying Index (QI) of an individual nutrient $q$ in the country $k$ and $N_q$ is the number of qualifying nutrients considered (=25). The

QI is defined as the ratio of a particular nutrient's amount contained in 2000 kcal of a given food/meal/diet relative to the Reference Daily Intakes[62] for those nutrients:[20]

$$QI_{q,k} = \frac{2000}{E_k} \cdot \frac{a_{q,k}}{DRI_q} \qquad (2)$$

Here, $E_k$ is the total daily caloric intake for country $k$ (in kcal per capita per day) and $a_{q,k}$ is the daily intake amount of a qualifying food nutrient $q$ in country $k$ (in g per capita per day). If the QI for a given nutrient is > 1, then the value is truncated to 1 on the rationale that once the requirement for a specific qualifying nutrient is met ($QI_{q,k} = 1.0$) any further provision of that nutrient does not serve additional nutrition function. It also guards against obtaining high nutrition balance (NB) score for diets containing abnormally high amounts of few individual nutrients and very low amounts of other. An NB score of 100 is achieved a diet satisfies at least 100% of the daily dietary requirement for every qualifying nutrient. Conversely, a value of 0 implies that none of the qualifying nutrients are contained in the diet.

It is noteworthy that we chose 2000 kcal in the numerator in Eq. 2 above, in order to be consistent with formula proposed by Fern et al.[20], although other values based on country-specific guidelines can also be chosen. Although suggested by Fern et al.[20] we could not include "Water" as a qualifying nutrient here because its amount in national food supply (e.g., tap and bottled water intake per capita per day) is not available from FAO's FBS.

**Disqualifying Nutrient Score**. For each country $k$, we first calculated the DI[20] for each of the four public health-sensitive nutrients $d$ (sugar, cholesterol, saturated fat, and total fat) as the ratio of their amounts contained in 2000 kcal of a given country's diet, and their MRV for those nutrients[62]:

$$DI_{d,k} = 100 \cdot \left( \frac{2000}{E_k} \cdot \frac{a_{d,k}}{MRV_d} \right) \qquad (3)$$

Here, $E_k$ is the total daily caloric intake for country $k$ (in kcal per capita per day) and $a_{q,k}$ is the daily intake amount of a health-sensitive food nutrient $d$ in country $k$ (in g per capita per day). If the DI value is > 100, the diet is deemed "compromised," because it contains disqualifying nutrients in values higher than the MRV relative to the total calories supplied by the diet[20]. The DI values of four nutrients were averaged to calculate a single DI score for each country. If the single DI score for a country came > 100, its value was truncated to 100. Finally, we derived the DNS (0 < DNS < 100), by simply subtracting the DI score obtained above from 100. Countries with higher intake of these four nutrients will have a lower DNS.

$$DNS_k = 100 \cdot \left[ 1 - \left( \frac{\sum_{d=1}^{4} DI_{d,k}}{N_d} \right) \right] \qquad (4)$$

Unlike Fern et al.[20], we did not include *trans*-fatty acid in DNS calculations, because its amount in different food items was not available in the USDA SR28 excel database[22]. We did not include "sodium" as a disqualifying nutrient since it is often added during processing/cooking in the form of salt. The intake amounts of sodium calculated from FAO's FBS[23] might thus be underestimates. For example, the mean sodium intake in the UK estimated from FBS is around 1.2 g per capita per day, which is around three times less than the mean value of 3.2 g per capita per day (equivalent to 8 g per capita per day of salt) estimated from 24 h urine samples in the 2014 National Diet and Nutrition Survey (https://www.gov.uk/government/statistics/national-diet-and-nutrition-survey-assessment-of-dietary-sodium-in-adults-in-england-2014).

One may argue our selection of a large number of nutrients to carry out the nutrition NB and DNS calculations owing to potential redundancies (e.g., including both cholesterol and total fat in DNS). Choice and selection of nutrients also vary across national guidelines. This is acknowledged by Fern et al.[20] who mentioned that past studies[35,63] have used anywhere from 5 to 23 nutrients; while calculating the nutrient density of foods (providing almost similar results) and the NB, DNS calculations can also be based on a smaller number of nutrients. Clearly, globally harmonized and consensus-based set of guidelines are needed on selection of particular nutrients to calculate the nutritional quality assessment of diets. For this study, we simply followed Fern et al.[20] in selecting the 25 qualifying nutrients for NB score and four disqualifying nutrients for DNS. The main criteria in selecting these particular nutrients was whether their daily required intake values have been published by the Institute of Medicine, National Academy of Sciences, and whether the nutrient compositional data for them existed in current databases such as USDA SR[22]. The detailed justification for the choice of these nutrients can be found in Fern et al.[20].

**Population Share with Adequate Nutrients**. We applied the estimated average requirement "cut-point" (EAR-CP) method to estimate the % of people in a given country with intakes of a particular nutrient above a demographically weighted requirement threshold for that country, termed as WtdEAR[21]. A normal

distribution of daily per capita nutrient intake amounts is first constructed for each nutrient around the mean by applying a coefficient of variation (CV) typically used for that nutrient. We could only include 17 nutrients in the analysis for which the EAR value is available. The CV values for each nutrient were obtained from Arsenault et al.[57] who carried out similar analysis for eight nutrients in three countries (Bangladesh, Senegal, and Cameroon). The CV for rest of the nine nutrients was assumed to be 25%[64]. The EAR of each nutrient was available for a given age and gender group[65] and was converted to single WtdEAR based on the population size in each age and gender group for each country in the year 2011[66]. Finally, we apply the EAR-CP method[21] to estimate the % of a country's population with intakes of each nutrient above the WtdEAR[21]. This indicator was then specified as the simple average of these population shares across 17 nutrients for a particular country. We did not account for bioavailability of different nutrients. For example, the absorption of zinc is reduced by phytate that chelates with zinc. As humans cannot digest phytate, the chelated zinc cannot be utilized[67].

**Ecosystem Stability indicators**. For estimating GHG emissions associated with national average daily diet, we linked intake amounts of 94 food items (g per capita per day) to the mean, minimum and maximum GHG emission factor values (g $CO_2$ eq. $g^{-1}$) obtained from Clune et al.[25]. This is the most recent comprehensive meta-analysis of its kind to date, reviewing 369 published life cycle studies and providing 1718 GHG emission factors for 168 food items.

To enable consistent comparison, Clune et al.[25] converted all collected emission factors from different studies to a common system boundary of a "regional distribution center (wholesale market)." Thus, emissions from "packaging" and "transport to regional distribution center" stages (also obtained through published literature) were added to studies where the system boundary finished at the farm gate. Emissions in the later life cycle stages of the food items such as during consumer travel to shops, food storage, cooking, and disposal were not included in the Clune et al.[25] values. For a few FAO food items (e.g., yams and coffee) for which GHG emission factors are not provided by Clune et al.[25], we used other sources[10,35,63,68].

For calculating the national blue water footprint, we follow the approach followed by Vanham et al.[8]. We first import the global average blue water footprints (l $g^{-1}$) of 94 FAO food items available from Mekonnen and Hoekstra[26,27] and multiply them with their respective intake amounts (in g per capita per day) in each country for the year 2011. Summing these up provided the diet related national average daily blue water footprint (in liters per capita per day).

The percentage of non-renewable energy use at the national level was obtained from data made available by the World Bank[59]. Due to lack of sector-specific data, we assumed that the % of non-renewable energy use in food systems of a country is the same as for its overall economy.

We used the "Environmental Performance Index (EPI)" as a proxy for food system indicator of "Ecosystem Status"[18]. The EPI ranks the performance of different countries in two broad policy areas (ecosystem protection and human health risk from environmental harm) through a suite of indicators[34]. We took a simple average of the following five indicators of ecosystem protection (each already available at 0–100 scale) relevant to food systems to calculate ecosystem status indicator for this study: Agriculture, Fisheries, Water Resources, Forests, and Biodiversity/Habitat.

The per capita land-use footprint (ha per capita) associated with each country's daily average diet in the year 2011 was obtained from Figure S1 of Alexander et al.[9]. They first derived crop and pasture areas (ha $kg^{-1}$) associated with 90 food commodities (50 primary, 32 derived commodities from them, and 8 livestock products) consumed in each country through FAO's country-level data on crop areas, yields, commodity uses and nutrient values[23]. Next they linked it with FAO's FBS data on national diets containing amount of each commodity consumed in each country (g per capita per day) to obtain the per capita land use footprint (ha per capita) of each country (crop and pasture combined).

The per capita biodiversity footprint (projected regional species extinctions per capita) of each country were obtained from Chaudhary and Kastner[28] who calculated the number of species (of mammals, birds, amphibians, and reptiles) projected to go extinct due to land use associated with total food consumption (considering domestic + imports – exports) of each country in the year 2011. They calculated this by combining the countryside species–area relationship model with global agriculture land use, crop yield maps, and FAO country-level data on food production and trade[23]. We used species richness decline as an indicator for which national footprint data was currently available[28] but future studies can include additional indicators of biodiversity loss such as loss of evolutionary history (phylogenetic diversity)[69]. It is noteworthy that all results are based on national consumption rather than national production (i.e., taking into account of where the foods were produced and attributing the impacts to the importing country).

The carbon, blue water, land, and biodiversity footprints derived above were converted to 0–100 scale as proposed by Gustafson et al.[18].

$$Indicator_i = 100 \times e^{[\ln(0.5) \times (F_i/F_{50})]} \qquad (5)$$

where $F_i$ is the footprint (e.g., carbon, water, land, and biodiversity) for the $i^{th}$ country under consideration, and $F_{50}$ is the global median value of this footprint across all 156 countries for the year 2011[18]. Using this equation delivers an indicator score of 100 for the case of zero footprint, a score of 50 for footprint equal

to global median and asymptotically approaching a score of 0 as footprint increases. It is worth noting that some indicators such as Gini index[60] or food availability score[37] per country are already available on a 0–100 scale, and thus were used as such here without any transformation (see Gustafson et al.[18] for details on each indicator).

**Alternative dietary scenarios**. We constructed three alternative dietary scenarios (HGDs, VGT, and VGN) for all 156 countries for the year 2011, following the procedure outlined by Springmann et al.[11] For each country, we calculated the intake of 94 food commodities (g per capita per day) under the three scenarios by adjusting their current diets for the year 2011, while maintaining the regional character of food consumption. A detailed discussion and rationale for different assumptions and constraints applied under these three scenarios are provided by Springmann et al.[11] and are briefly summarized here.

The HGD alternative scenario assumes that people consume just enough calories to maintain a healthy body weight[32] and the food intake amounts comply with the global dietary guidelines on healthy eating[30,31]. The four constraints included for constructing a HGD diet (per day) areas follows: fewer than 50 g of sugar,[31] 43 g of red meat,[30] a minimum of five portions (400 g) of fruits and vegetables,[31] and 2200–2300 kcal of total energy intake (depending on the age and sex composition of the population)[32]. For simplification, we assume the optimal total energy intake under the three alternative scenarios to be 2300 kcal per capita per day.

The five constraints applied in the VGT scenario (per day) are as follows: fewer than 50 g of sugar, no red meat or poultry or fish, 2300 kcal of total energy intake, at least six portions of fruits and vegetables (6 × 80 g), and at least one serving (80 g) of legumes/pulses[5,11,29]. The constraints in VGN scenario were the same as VGT, except a minimum seven portions of fruits and vegetables per day (7 × 80 g) instead of six and also no eggs or dairy. It is noteworthy that there is no constraint or allowable level defined for eggs/dairy intake in the VGT scenario and it is treated in the same way as other food items such as wheat or rice. Thus, if the current caloric intake of a country is 2300 kcal per capita per day, then the amount of egg or dairy intake levels remain unchanged from reference to VGT scenario.

Intake amounts (g per day) of food items (e.g., vegetables, fruits, and pulses) in a country that currently consumes less than the minimum recommended amounts were adjusted upwards to meet the minimum requirement, whilst intake amounts of items (e.g., sugar, red meat, poultry, fish, eggs, or dairy) which exceeded the maximum recommended intake were adjusted downward to meet that ceiling. Total energy intake in each scenario was adjusted to the recommended 2300 kcal levels by varying the country-specific consumption amounts of staple foods (grains, roots, and pulses), which preserved the regional character of food consumption[11].

One novelty of our analysis is that although Springmann et al.[11] considered 16 broad aggregated food groups, we constructed the dietary scenarios maintaining all 94 food items of FAO's FBS[23]. This is because the nutrients such as vitamins A, C, or folate can vary a lot from one individual commodity to the other within the same broad food groups such as fruits, vegetables, and pulses.

Finally, following the methods described above, we recalculated the three nutritional (NBS, DNS, and PANs) and two environmental (carbon footprint and water footprint) indicators under these scenarios for each country.

**Data availability**. All input data and the procedure for deriving the seven food system metrics and 25 indicators are publicly available through the references and online sources cited in Table 1 and Methods described above. Intermediate and final results are provided in supplementary excel file accompanying the paper as Table 1–7.

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

## Acknowledgements

The research was partially funded by the National Research Programme "Sustainable Economy: resource-friendly, future-oriented, innovative" (NRP 73) by the Swiss National Science Foundation (Grant number: 407340_172415).

## Author contributions

A.C. conceived the idea and designed study, carried out all data compilation, processing, and calculations for nutritional and environmental indicators, and analyzed results and wrote the first draft. D.G. compiled the economic and social metric scores, and contributed in writing. Both D.G. and A.M. provided comments on the draft.

## Additional information

**Competing interests:** The authors declare no competing interests.

