## [Peer Review File · Nature Communications]

Reviewers' comments:

Reviewer #1 (Remarks to the Author):

This manuscript makes a useful contribution to existing studies on global food systems sustainability through the combination of disparate nutritional, social, and environmental indicators. There is considerable need to better understand the synergies and trade-offs involved in achieving multiple targets with respect to specific indicators for policy and planning. By considering how multiple indicators might change under contrasting diet scenarios, the findings presented here offer considerable new insight (Table 2 is very interesting, and offers much potential for deeper analysis/visualization).

For the most part, I find the manuscript to be well-written and the methodology seems appropriate at this scale of analysis (e.g., the quite challenging 'decomposition' of the FAO Food Balance Sheets for further analysis of individual food items across countries and linkage to various nutritional metrics, etc.). Below, I outline a series of more substantive and more minor concerns, which I hope the authors will carefully consider in any revision of this manuscript.

One of the most specific and important conclusions in the manuscript is given at lines 99-101 and lines 399-404, in that a transition toward more plant-based diets could have major implications for improving environmental and health dimensions of food systems, but that it may need to be accompanied by micronutrient supplementation. Of course, this has been discussed to some degree elsewhere, but it is very clearly illustrated in this paper and juxtaposed in a novel and interesting way with a series of indicators and alternative diet scenarios. I do feel that there is a bit more room to visualize and statistically assess these synergies and trade-offs, perhaps even just with simple scatterplots, but the use of choropleth maps and tables makes a good first attempt. The alternative scenario results shown in Table 2 are really quite exceptional. It would be quite interesting to see these in more detail, including for individual countries.

****Substantive comments****

Choice of indicators: I realize that the choice of indicators builds out of previous work by Gustafson et al., but this point was one of the most challenging for me in interpreting the study's findings. On one hand, some indicators are directly related to food and nutrition security, whereas some are rather more indirectly related—but this seemed to not be discussed as much as I was expecting, that is, what do we really learn from any given indicator and different combinations of indicators? There are so many indicators to choose from, that I think there could be some further discussion of why these indicators are the most useful. What about using the Human Development Index (HDI)? Or IFPRI's Global Hunger Index? Or other considerations such as food price volatility for food system resilience? In this respect, it is also important to acknowledge where there is overlap (or not) among indicators. I found the use of correlation with GDP per capita to be very useful, but was left asking: How correlated are the indicators with each other (e.g., PAN, DNS, and NBS)? A simple scatterplot matrix could help to elucidate that. Finally, with respect to the per capita land use indicator, I would strongly encourage the authors to consider separating out pasture and cropland for a 'fairer' comparison among countries (e.g., lines 182-184 show Mongolia and Namibia – but these are very extensive grazing systems – how does that compare to the land use underlying a Western diet?).

The manuscript is clearly written, but it could be a bit more concise—leaving room for deeper discussion of the actual results. This could easily be addressed by removing some of the methods, as there are essentially two methods sections in the manuscript at present. Lines 64-88 could likely be shortened – with critical info not covered in the Methods moved there and any info that is covered twice, deleted.

The authors might consider making it as clear as possible what can and cannot be learned from

interpretation of nation-scale indicators. There was not much acknowledgement of subnational variation in food systems issues, such as food insecurity. One of the only prominent mentions of these subnational patterns and inequities comes in the Methods at lines 486-490, where it is mostly about limitations in household surveys.

Scaling: the very interesting alternative diet scenario results shown in Table 2 are perhaps the case where the (changes in) absolute values are most meaningful to provide benchmarks for change comparison. One point of confusion for me was how the indicators were scaled for use outside the scenario analysis. The Methods section helped to clear this up – my understanding is that several metrics were calculated within a scale of 0-100, whereas others (e.g., the carbon and blue water footprints) were scaled by using Equation 5 (line 170). However, if one thinks of an indicator such as Gini coefficient, where 0 and 100 correspond to perfect (in)equality, would it not perhaps be fairer to use the order in the actual country data, i.e., so that the data are scaled so that the countries with the highest/lowest Gini correspond to 0 and 100, respectively? This simple relative ranking or scaling seems like one alternative to consider country unitless country comparisons across multiple indicators, with less sensitivity to actual distribution of data for particular indicators.

****Other minor considerations****

Abstract, line 8: for environmental scientists reading this paper, there might be some confusion about the use of the term 'nutrient' (i.e., crop fertilizers versus food nutrition). Can the word 'nutrient' be changed to 'nutrition' throughout, or perhaps just change to 'food nutrients'?

Abstract, lines 12-13: the phrasing is a little odd, perhaps change to "replacing animal protein with plant-based proteins", or something like this?

Introduction, lines 23-24: while a "growing world population" is certainly important to highlight, I wonder why the authors would not also take a few words to acknowledge that changing diets and consumption patterns are placing even more pressure on our food systems than population growth.

Line 39: the "etc." seems to be overkill here given the breadth of the terms used in this sentence. I actually wrote "be more specific" in my initial read of the manuscript, thinking that another sentence going into a bit more detail would be helpful to readers to flush out what these multiple domains represent.

Lines 42-43: It seems somewhat inappropriate to say "a limited number of global studies" – there have actually been quite a large number of global analyses pointing to the impacts of consuming animal products on GHG emissions and resource footprints. What is rarer is the link of these environmental dimension to health/nutrition dimensions, where work by, as just one example, David Tilman and colleagues stands out as one of only a handful that bridge multiple dimensions. I wonder if the authors might be able to elucidate these tradeoffs and synergies a little more.

Lines 80, 164, and 170: the term "emissions factor" is appropriate to carbon, but not blue water. Similarly, it might be easier to say "blue water consumption" rather than "net water withdrawals", for clarity. The water footprint addresses water consumption (evaporation or incorporation into a product), not withdrawals. At line 170, likewise, it might be more appropriate to change "withdrawals" to "consumption".

Lines 193-195: this statement that "The higher nutritional quality of most nations' diets comes at the cost of higher damage to the environment..." seems rather contentious and a little too bold based on the actual results depicted in this study. Can the authors temper this a bit based on the actual study findings?

Lines 309-312: the statement about small differences in blue water requirements of plant- and animal-based foods is surprising and counterintuitive. The authors should carefully check this. It is also contradicted by the result described at lines 385-388 about a sharp drop in freshwater consumption under the VGN scenario.

Line 349: very small nuance, but the authors might say “anthropogenic land used for agriculture” instead of ‘land to produce food’ in terms of land clearing and its biodiversity impacts. Much of the recent land expansion in the tropics has been linked to commodity crops, such as soybean, which are indirectly consumed in foods after being fed to livestock. This is one of the cruxes of the challenge around inefficiencies in our food systems.

Lines 365-370: the current discussion of gender equity and per capita incomes seems rather simplistic. One of many potential references that could help address the issue of gender equity and social justice with respect to food systems and economic development is Schipanski et al. (2016), or a reference on gender equity and social justice therein.

--Schipanski, M.E., MacDonald, G.K., Rosenzweig, S., Chappell, M.J., Bennett, E.M., Kerr, R.B., Blesh, J., Crews, T., Drinkwater, L., Lundgren, J.G. and Schnarr, C., 2016. Realizing resilient food systems. *BioScience*, 66(7), pp.600-610.

Lines 393-394: this represents an increase in per capita GHG emissions from currently low levels in a relative sense.

Lines 588-591: Can the authors be more specific about the quantity of eggs/dairy ‘allowed’ in the VGT scenario, or at least how these food items are handled?

Table 1: please consider changing the label for Shannon Diversity to “Shannon Diversity of Consumption”, or something indicating diversity of food supply, to more clearly separate from Food Production Diversity.

Reviewer #2 (Remarks to the Author):

The authors present an ambitious effort to characterize the sustainability of the global food system. The manuscript builds on several related efforts that proposed either proposed indicators or presented related results. In addition, the authors present a few new indicators. It's clear that a diverse set of expertise worked on this paper.

Although there is a wealth of information, the main messages are not very clear. Very little is done to integrate the findings into a few clear messages. Do the results mean anything beyond presented the current status and showing that there's lots of variation around the world? What's new here—does the main message differ from other reports or is the main progress that the indicators are quantified? Where there any surprises?

The text needs a lot of editing to have the main messages be distinguished from the details. It currently reads as a group edited, indicator-by-indicator prose in the methods, results and discussion sections. It should be a bit more formal as well, changing phrases that are informal such as “score well on most fronts” and “stood close to”

It was challenging for me to comment on all aspects of the manuscript as it covers (and requires) such a broad set of expertise. However, for all indicators, the authors cited references that demonstrated that they used accepted methods to compute the metrics. These are summarized in Table 1 as well as throughout the text. They also acknowledged shortcomings of the approaches and data limitations.

More specific comments:

Given the uncertainty in the values for the computed metrics, can you really use the results to “set quantitative goals (e.g. reducing per capita GHG emissions below the threshold level of 2.055 kg CO₂ eq).”? Elsewhere the authors state that “We found that the carbon footprint values per country have high uncertainty, varying by over an order of magnitude.” Similar uncertainties will arise in estimates of food production in some countries if relying on FAO data. As a result, there will be high uncertainty in all the nutrition metrics. Further, while aggregating indicators to a 0-100 scale makes it easier to compare across aspects of the food system, but it also indirectly assumes that all the indicators are equally weighted and the factors (environmental, nutrition, etc) are independent. Obviously neither assumption is true. Are these main results or national level metrics better used to identify priorities and track directional progress rather than setting specific goals? I agree that the authors approach is the best that can be taken given the data limitations. I just don’t think the utility should be oversold to the point of setting or measuring progress toward quantified goals with any high level of precision.

There is little comparison of the results with previous studies. Although other studies have not addressed the full set of metrics as the work here, there has been substantial work on several of the specific issues. These include malnutrition/stunting and GDP, diets and GHGs, water and diets, etc.

Table 2 is difficult to interpret without the current values. I suggest that the current diet/consumption numbers are added to the chart. That would put the change values into context. However, it still does not address the problem of the metric values being intuitive as have no idea if a change of 5 is a lot or little. Is there a better way to present the changes so that they are more intuitive? Further, it’s helpful to have the regional summaries here, similar to Figure 4. But how were the regional values calculated? Are they per capita, average of country values, etc.?

Title: I don’t have any immediate suggestions for changes, but I generally don’t think of nutrition and food safety when I think of sustainability. Also, why the plural for systems? Is it because the analysis is at the country scale or because of the scenarios?

The scope of the paper is very broad, but should address a few key items in the discussion. For example, climate change is only referenced in terms of GHG emissions and resilience. How will a warming world affect the indicator values? Do the trends in population and diet add constraints? Over a third of all calories are traded—could many of the problems in the global food system be fixed by trade?

Resilience - crop diversity may be a good indicator, but other factors are likely more important in most places. These include: irrigation, nutrient management, soil organic matter, and crop selection.

Is one of the main solutions to reduce income and gender inequality? Are there places where gender and income equality measure are high but not adequate food and nutrition? Related, the income and gender inequality metrics provide insights into sub-national variations. Would your main results be different if other indicators were sub-national? Or do you think the main patterns would be the same?

Diversity indices such as the Shannon index are nice for aggregating information. But the metric is very abstract and there are different pathways for getting similar index values. How does this limit the indicators utility for guiding action?

-Paul West

Reviewer #3 (Remarks to the Author):

This is an excellent and important paper. The data is well-presented, with appropriate caution. The results are of great policy significance. The style is clear and readable. I strongly support publication of this paper with no significant amendments. The only caution is that I am not a professional nutritionist so their critical review comments may be more important.

What they did

They present the first global scale analysis quantifying the status of national food system performance of 156 countries, employing 25 sustainability indicators across seven domains. Their nation-specific quantitative results will help policy-makers to set improvement targets on specific areas to improve nutritional intake and sustainability.

Our biggest health and nutrition challenge is to provide a balanced diet to a growing world population in a sustainable manner. Their seven metrics of sustainable nutrition security are (1) Food Nutrient Adequacy; (2) Ecosystem Stability; (3) Food Affordability and Availability; (4) Sociocultural Wellbeing; (5) Food Safety; (6) Resilience; and (7) Waste and Loss Reduction (see Table 1). Each of the metrics comprises multiple indicators that are combined to derive an overall score (0–100). They quantify the food system sustainability status of 156 countries for the year 2011 using the seven metrics and 23 underlying with two additional indicators on biodiversity impacts and health sensitive nutrient intake. They also calculate a new indicator, the Disqualifying Nutrient Score (DNS), by comparing the total daily intake of five public health sensitive nutrients (sugar, sodium, cholesterol, saturated fat and total fat) with their Maximal Reference Values (MRV). Countries with lower intake of these sensitive nutrients score higher on this indicator. They also calculate the Population share with Adequate Nutrients (PAN), the Ecosystem Stability metric, and the land and biodiversity footprint of each country's daily food consumption. Finally they constructed three alternative dietary scenarios [healthy global diets (HGD), lacto-ovo vegetarian (VGT), and vegan (VGN)] based on global dietary guidelines on healthy eating and recommended levels of fruits, vegetables, red meat, sugar and total caloric intake for each of the 156 countries.

Limitations of the data

I'm a paediatrician working in policy and practice but not a professional nutritionist. The authors present a good narrative in the discussion about uncertainties and limitations in interpreting the results which seems clear and reasonable. For example, the FBS provides information on only 94 food items. They used nutrient values of different food items from a single food composition database, and report that carbon footprint values per country have high uncertainty. They present good ideas for future studies. For example, quantifying the impact of food systems through additional indicators (e.g. nitrogen and phosphorus emissions causing eutrophication and acidification), and developing more sophisticated scenarios with constraints on other food groups, considering consumer preferences, with additional nutritional, environmental, economic, and social indicators of sustainability, are important areas for future research.

Key results that caught my eye as highly relevant for policy

1. Income A positive effect of higher income on national nutritional diversity and adequacy (Table 1). However, there was a strong negative correlation between DNS and GDP.
2. Ecosystem Stability: Carbon footprint varies from around 0.7 kg CO₂eq capita⁻¹ day⁻¹ for African countries such as Mozambique, Ethiopia and Malawi to >4 kg CO₂ eq. capita⁻¹ day⁻¹ for New Zealand, Australia, USA, France, Austria, Argentina and Brazil. Per capita food related GHG emissions with "sustainable levels," at no more than 50% of the yearly total per capita emissions shows 81 of 156 countries (representing ~52% of global population) have GHG emissions above this threshold. Bovine meat consumption contributes the most to the GHG emissions. "Blue water footprint" is similar to carbon footprints.
3. Women's empowerment. A very weak correlation with GDP is Gender Equity which directs policymakers to pay more attention to, and invest more, in women's empowerment. Gender equity remains a big problem, suggesting that women's empowerment concerns persist even as national incomes rise.
4. Waste and Loss Reduction Higher-income countries generally waste much larger percentages of their food at the post-consumer level than do lower-income countries. However, pre-consumer

losses in lower-income countries are relatively large. They measured the produced food that is not either lost (pre-consumer) or wasted (post-consumer) in a country. High-income nations such as Canada, USA, Australia and EU countries score lower (~60) than low or lower-middle income nations

5. Alternative dietary scenario outcomes

Dietary changes toward fewer animal and more plant-based foods result in significant reductions in daily per capita disqualifying nutrient intake (e.g. >50 point increase in DNS for North America and Europe) but relatively small improvements in the Nutrient Balance Score. They found that adopting diets low in animal sourced foods can significantly reduce the food related per capita GHG emissions and freshwater use of nations (e.g. >70% and 40% reductions, Adoption of alternative diets (healthy global diets (HGD), lacto-ovo vegetarian (VGT), and vegan (VGN)) leads to high improvement in Disqualifying Nutrient Score (DNS) across Europe, Latin America and North America. Also they would result in food related per capita GHG emissions 12, 40 and 50% lower than under current diets respectively, and where at present 81 out of 156 countries exceed the sustainable threshold level (2.055 kg CO₂ eq. capita⁻¹ day⁻¹), the number of countries not meeting this criterion drops to 73, 8 and just 7 under the HGD, VGT and VGN, respectively.

6. Diet in the wealthy west

Dietary changes in North America and Europe lead to the largest reductions in per capita carbon footprint primarily because of a decrease in total caloric intake and reduced meat intake levels under the three scenarios. The average intake of five health sensitive nutrients: sugar, sodium, saturated fat, total fat and cholesterol is currently 2, 5, 2, 2 and 7 times higher for high-income nations than the low-income nations. Policymakers can focus on increased production of particular crops, dietary diversification, cut food imports, biofortification, tax incentives and so on.

7. Food and biodiversity The current rate of species extinction is over 100 times the background rate, so anthropogenic land use to produce food is one of the major drivers of global biodiversity loss, and the land footprint is not a good proxy for the biodiversity footprint.

Reviewer #4 (Remarks to the Author):

The paper proposes an analysis of the sustainability of national food systems of 156 countries using 25 sustainability indicators defined by Gustavson et al. and available global data.

It is a useful enterprise, that confirms more partial analysis. It also brings forward some interesting points, like the fact that the higher nutritional quality of diets can go against to the environment (193-195), a fact that is often not considered in global papers that insist on environment and health go hand in hand. This point would deserve to be better emphasized, including in the discussion.

Given that this paper is likely to be of interest for a very diverse set of researchers it would benefit from being clearer on some points, in particular on the way some indicators are calculated or applied.

For instance it seems that the indicators are calculated/applied to the consuming country; it needs to be said. It would also be good to recall, line 33-34 that consumption and production spaces are now quite different, especially for some products that are widely traded. In other words some of the impacts determined by consumption are different depending on the place/country of production.

Some of the indicators, like the environmental performance index are not determined only by food systems (mainly but not only), it should be said. For other, in

particular gender equity it is not clear if its scope is the food system or the whole country. This is important given the prominence of women labor at most stages of the food value chain.

Precisely because some indicators seem to be country wide and not only national food system wide, it would be good to precise for instance line 140 that it is the carbon foot print of the diet. same for other indicators for which we are used to see per capita economy wide values like the water footprint.

For some indicators, like for food and losses for instance, the data used has not been constructed to be downscaled at national level and thus loses certainty when downscaled (for a short discussion on the methodology used and its accuracy see HLPE (2014) Food losses and waste in the context of sustainable food systems. A report by the High Level Panel of Experts on Food Security and Nutrition of the Committee on World Food Security. Rome: FAO. Box 1 p28)

The biggest issue however concerns the economic and social dimensions. There is no indicator of the economic (and social) sustainability of the food system itself. a point which was already noted in Gustafson et al, but without taking the full measure of its consequences. In fact with the main economic indicator being affordability it seems that sustainability would be the cheapest food possible which does not account neither for the fact that most hungry and malnourished are actually food producers, and that there could be no long term sustainability to the sector without enough investment, returns on investment nor attractiveness for young workers without decent income. Even if it may not be easy to define such an indicator (of economic health of the system) it is a major gap that absolutely needs to be highlighted.

The discussion on the alternative scenario outcomes should also give room to potential outcomes of substitutions, for environmental impacts (potential increase of GHG emissions due to greenhouses in some colder countries when vegetables would substitute ASF (see for instance Tom M S, Fischbeck P S & Hendrickson CT (2015) Energy use, blue water footprint, and greenhouse gas emissions for current food consumption patterns and dietary recommendations in the US. *Envt Syst Dec* 1–12.) And importantly, the lack of proper indicator on the economic and social sustainability of food systems does not enable even to consider potential impacts on farmers, and especially on small farmers of a reduction of ASF consumption, livestock representing in many countries a critical way to increase economic productivity of both land a labor and to escape poverty.

More detailed comments:

25-26, no mention at all of economic issues. food systems are by far the biggest employer in the world, particularly in poor countries, and they also represent an important part of GDP, not only in developing countries, but also in developed countries with a different distribution of income generated between agriculture/transformation/distribution.

182-183 extensive grazing is also linked to very specific ecosystems and species, that are in fact preserved by this mode of land use. It should be mentioned here. As rightly said later (350) the land footprint is not a good proxy for the biodiversity footprint.

223, as explained above, you cannot say that the previous indicators dealt with economic considerations, rather economic access for consumers.

279-270, propose to write rather "of disqualifying nutrients" and to delete sugar from the parenthesis as the link between meat consumption and sugar intake seems dubious.

380 add "improving " before "food safety".

methods: you may want to complete this section as the method is missing for some of the

indicators, or at least give an easily reference to be consulted.

Reviewer #5 (Remarks to the Author):

This is a thorough report of an analysis of the sustainability of food systems in 156 countries, which develops metrics across multiple indicators to assess the current nutritional status and environmental impact of food production and consumption. The authors should be commended for their hard work bringing multiple metrics together in a single place. The holistic nature of the assessment of the impact of the food system is the biggest strength of this paper.

Major comments:

Page 4, lines 74-76: "The national daily average intake of each nutrient was obtained by combining a food composition database with the FAO food balance sheet (FBS) database." What about nutrients added during processing? This approach will give very little information about sodium levels in food. Indeed, Table S2 suggests that sodium levels have been greatly underestimated by their approach. The region with the highest level of sodium consumption is North America (1216mg capita-1 day-1), but this level is far lower than the MRV of 2400mg (equivalent to 6g of salt). But mean consumption of salt in the UK (for example), estimated by 24 hour urine collection (the gold standard), is about 8 g day-1 (estimates from the 2014 National Diet and Nutrition Survey) - far higher than levels estimated for this analysis. I suggest that the authors should remove sodium from their list of disqualifying nutrients.

The authors do not mention the Global Dietary Database (GDD), which combines FBS estimates with population surveys of dietary intake in order to model comparable estimates of food and nutrient intake at a country level. The GDD approach overcomes many of the limitations of using FBS estimates and this analysis would have benefited from using data from the GDD (which may not have been possible). At the very least, the authors should comment on the GDD when discussing the limitations of using FBS data for characterising food and nutrient intake.

What is the justification for the choice of nutrients for both the NBS and the DNS? I am surprised to see both cholesterol and total fat included in the DNS.

Estimating the blue water impact of each country: was this done with the region-specific results from Mekonnen & Hoekstra? If so, how did the authors gain country-level estimates from the sub-country regional estimates reported in Mekonnen & Hoekstra?

Presumably all of the results are based on national consumption rather than national production (i.e. taking account of where foods were produced (and importing environmental footprints to the importing country)? This should be explicitly stated.

The discussion does not provide much insight. It generally just restates the results and states that these may be important for setting country-specific strategies for addressing the sustainability of food systems. Lines 317-404 could be shortened considerably.

Minor comments:

Page 26, equation 2: Please can you give definitions of Ek and aq,k

Page 7, lines 119-122: "we found that the global median value was 75.73% (averaged over 17 essential nutrients) meaning a significant share of world population (ca. 25%) is not receiving adequate nutrients currently (Table 1)" This statement only makes sense if the global median is calculated with weighting for different populations between countries. Is that the case? Even if it is, I still don't think the 25% figure makes much sense, as the 75.73% figure is a median of

means of 17 micronutrients. So if a micronutrient was added for which everyone had sufficient consumption, then all of the country-level means would be adjusted upwards and the global median would be adjusted upwards, which would suggest that a smaller proportion of the global population was not receiving adequate nutrition, even though no change has been made to the underlying data on the 17 micronutrients currently included. In summary, I suggest that the statement that 25% of the global population do not receive adequate nutrition should be dropped!

Reviewer #1 (Remarks to the Author):

Comment#1: This manuscript makes a useful contribution to existing studies on global food systems sustainability through the combination of disparate nutritional, social, and environmental indicators. There is considerable need to better understand the synergies and trade-offs involved in achieving multiple targets with respect to specific indicators for policy and planning. By considering how multiple indicators might change under contrasting diet scenarios, the findings presented here offer considerable new insight (Table 2 is very interesting, and offers much potential for deeper analysis/visualization). For the most part, I find the manuscript to be well-written and the methodology seems appropriate at this scale of analysis (e.g., the quite challenging ‘decomposition’ of the FAO Food Balance Sheets for further analysis of individual food items across countries and linkage to various nutritional metrics, etc.). Below, I outline a series of more substantive and more minor concerns, which I hope the authors will carefully consider in any revision of this manuscript.

One of the most specific and important conclusions in the manuscript is given at lines 99-101 and lines 399-404, in that a transition toward more plant-based diets could have major implications for improving environmental and health dimensions of food systems, but that it may need to be accompanied by micronutrient supplementation. Of course, this has been discussed to some degree elsewhere, but it is very clearly illustrated in this paper and juxtaposed in a novel and interesting way with a series of indicators and alternative diet scenarios.

I do feel that there is a bit more room to visualize and statistically assess these synergies and trade-offs, perhaps even just with simple scatterplots, but the use of choropleth maps and tables makes a good first attempt.

The alternative scenario results shown in Table 2 are really quite exceptional. It would be quite interesting to see these in more detail, including for individual countries.

Response: Upon reviewer’s request, we now added Supplementary Table S6 showing the dietary change consequences on the five indicators for each of the 156 countries and three scenarios (HGD, VGT and VGN).

We agree with the reviewer that there is more room to further visualize and statistically assess the indicators. We tried best to convey key findings while staying within word limit in the main text and do provide all values in the Supplementary Information excel file so that a reader interested in a particular result can utilize it for further analysis in future. Below we address the substantive and minor comments by all five reviewers. We now include several brief lines in the main text addressing each of the issue raised by the reviewers and direct the reader to Supplementary Tables for further details.

****Substantive comments****

Comment#2a: *Choice of indicators:* I realize that the choice of indicators builds out of previous work by Gustafson et al., but this point was one of the most challenging for me in interpreting the study’s findings. On one hand, some indicators are directly related to food and nutrition security, whereas some are rather more indirectly related—but this seemed to not be discussed as much as I was expecting, that is, what do we really learn

from any given indicator and different combinations of indicators? There are so many indicators to choose from, that I think there could be some further discussion of why these indicators are the most useful. What about using the Human Development Index (HDI)? Or IFPRI's Global Hunger Index? Or other considerations such as food price volatility for food system resilience?

Response: We agree with the reviewer that many other indicators of sustainability exist in the literature. In our manuscript we already very briefly summarize the criterion for selection of these indicators in Introduction section lines 64-66:

"These metrics were selected through consensus building activities involving a number of nutrition, economic, food system, and climate change experts representing a range of global public and private institutions (Acharya et al. 2014)."

We cited the ILSI-RF report (Acharya et al. 2014) at line 66 that discusses the sustainable nutrition security assessment concept in detail for the readers interested in the approach. For example, during the consensus building activities, it was found that several other potential indicators were highly correlated with the existing 23 indicators and therefore were superfluous.

Further, we acknowledge that the list is not exhaustive and mentioned the rationale for selecting the seven metrics and associated indicators at Discussion section lines 473-481:

"Our list of 25 indicators is not exhaustive, but a key criterion in the selection of these indicators is that they can be derived from data that are either directly available for all countries (e.g. food affordability & availability)³⁶ or can be readily estimated by processing global databases such as FBS²³ (e.g. Nutrient Balance Score - see methods). This means that the indicator and metric scores can be calculated and regularly updated for subsequent years to track the progress of national food system sustainability. Future studies should extend our work by quantifying the impact of food systems through additional indicators (e.g. nitrogen and phosphorus emissions causing eutrophication and acidification) and comparing them to their planetary boundaries (sustainable levels)."

It was beyond the scope of this study to discuss the choice of indicators in detail. The main goal of this study was to apply previously developed quantitative indicators to each country and identify areas of improvements. As appropriate data become available, the list of indicators can be increased or modified. For example, we include two new indicators - the biodiversity footprint and disqualifying nutrition score (DNS) that were not included in Gustafson et al.

Comment#2b: In this respect, it is also important to acknowledge where there is overlap (or not) among indicators. I found the use of correlation with GDP per capita to be very useful, but was left asking: How correlated are the indicators with each other (e.g., PAN, DNS, and NBS)? A simple scatterplot matrix could help to elucidate that.

Response: Thanks for pointing this out. We did provide Spearman rank correlation coefficient matrix comprising all 25 x 25 indicator combinations in Supplementary Table S5 but did not mention this explicitly in the main text. We now mention this at Table 1

title itself (lines 128-129): “See Supplementary Table S5 for correlation value of all 25 × 25 indicator combinations.”

In addition, for nutrition indicators, we now added the following sentence at lines 149-150 to direct the reader to Table S5: “We found that all individual indicators are positively correlated with each other except DNS which is negatively correlated to the other five indicators (see Supplementary Table S5 for Spearman correlation coefficient between all indicators).”

For example, Table S5 shows that the correlation coefficient between PAN and NBS is 0.70, between PAN and DNS is -0.42 and between NBS and DNS is -0.28.

Comment#2c: Finally, with respect to the per capita land use indicator, I would strongly encourage the authors to consider separating out pasture and cropland for a ‘fairer’ comparison among countries (e.g., lines 182-184 show Mongolia and Namibia – but these are very extensive grazing systems – how does that compare to the land use underlying a Western diet?).

Response: We agree with the reviewer. While we keep the results on total land use per capita (Fig. 2c) as the total national land footprint, we now also report the per capita pasture and crop land use separately for each country. We now report these results in Supplementary Table S1 and mention it at lines 204-206:

“On average, 66% of the total land footprint of African and Latin American countries comprises of pasture land compared with just 33% for South Asian and 50% for East Asian and European countries (see Table S1 for the % of total land footprint due to pasture and crop land per country).”

Comment#3: The manuscript is clearly written, but it could be a bit more concise—leaving room for deeper discussion of the actual results. This could easily be addressed by removing some of the methods, as there are essentially two methods sections in the manuscript at present. Lines 64-88 could likely be shortened – with critical info not covered in the Methods moved there and any info that is covered twice, deleted.

Response: We thank the reviewer for this useful suggestion and have now deleted several lines in the introduction section that repeat the information in Methods section.

Comment#4: The authors might consider making it as clear as possible what can and cannot be learned from interpretation of nation-scale indicators. There was not much acknowledgement of subnational variation in food systems issues, such as food insecurity. One of the only prominent mentions of these subnational patterns and inequities comes in the Methods at lines 486-490, where it is mostly about limitations in household surveys.

Response: We now acknowledge this in discussion section lines 486-491:

“We could not account for the sub-national variation in food systems issues such as food insecurity and therefore acknowledge the limitation of our country-average indicator and metric scores in policy making. Indeed, the indicators would have much higher practical utility if reported at sub-national scales and separately for different age, sex and different socio-economic groups of the population. However, this would require significant data

collection efforts at much finer spatial resolution than currently available.”

Comment#5: *Scaling:* the very interesting alternative diet scenario results shown in Table 2 are perhaps the case where the (changes in) absolute values are most meaningful to provide benchmarks for change comparison.

One point of confusion for me was how the indicators were scaled for use outside the scenario analysis. The Methods section helped to clear this up – my understanding is that several metrics were calculated within a scale of 0-100, whereas others (e.g., the carbon and blue water footprints) were scaled by using Equation 5 (line 570). However, if one thinks of an indicator such as Gini coefficient, where 0 and 100 correspond to perfect (in)equality, would it not perhaps be fairer to use the order in the actual country data, i.e., so that the data are scaled so that the countries with the highest/lowest Gini correspond to 0 and 100, respectively? This simple relative ranking or scaling seems like one alternative to consider country comparisons across multiple indicators, with less sensitivity to actual distribution of data for particular indicators.

Response: We agree with the reviewer and now include the absolute values of indicators for current (REF) diets in Table 2 to provide more meaningful change comparison.

Regarding the scaling, we forgot to mention this in the main text and thank the reviewer for pointing this out. For indicators such as Gini coefficient, we did use the order in the actual country data and did not apply Eq. 5 to them.

We now clarify this in the Methods section lines 676-678:

“Note that some indicators such as Gini index³⁷ or food availability score³⁶ per country are already available on a 0-100 scale and thus were used as such here without any transformation (see Gustafson et al.¹⁸ for details on each indicator).”

****Other minor considerations****

Comment#6: Abstract, line 8: for environmental scientists reading this paper, there might be some confusion about the use of the term ‘nutrient’ (i.e., crop fertilizers versus food nutrition). Can the word ‘nutrient’ be changed to ‘nutrition’ throughout, or perhaps just change to ‘food nutrients’?

Response: Thanks for mentioning this. We now remove the word ‘nutrient’ to avoid confusion and instead use ‘nutrition’ or ‘food nutrients’ as appropriate throughout the text.

Comment#7: Abstract, lines 12-13: the phrasing is a little odd, perhaps change to “replacing animal protein with plant-based proteins”, or something like this?

Response: We now changed it as suggested by the reviewer above. Instead of saying ‘less meat and more plant-based foods’, we now say “animal protein with plant-based proteins” at line 13 in abstract.

Comment#8: Introduction, lines 23-24: while a “growing world population” is certainly

important to highlight, I wonder why the authors would not also take a few words to acknowledge that changing diets and consumption patterns are placing even more pressure on our food systems than population growth.

Response: We agree and now added following lines 24-26 to acknowledge above factors:

“Meeting the increasing demand for nutritious food in the face of growing world population, consumption levels, dietary shifts and the consequent environmental degradation, constitutes a major challenge for humanity in this century.”

Comment#9: Line 39: the “etc.” seems to be overkill here given the breadth of the terms used in this sentence. I actually wrote “be more specific” in my initial read of the manuscript, thinking that another sentence going into a bit more detail would be helpful to readers to flush out what these multiple domains represent.

Response: We agree and now removed “etc.” from this sentence.

Comment#10: Lines 42-43: It seems somewhat inappropriate to say “a limited number of global studies” – there have actually been quite a large number of global analyses pointing to the impacts of consuming animal products on GHG emissions and resource footprints. What is rarer is the link of these environmental dimension to health/nutrition dimensions, where work by, as just one example, David Tilman and colleagues stands out as one of only a handful that bridge multiple dimensions. I wonder if the authors might be able to elucidate these tradeoffs and synergies a little more.

Response: We agree and now remove the word “limited” from the sentence. We also add following lines to end this paragraph at lines 57-58:

“In sum, studies that are global in scale and evaluate food systems using multiple indicators of sustainability are rare.^{10,11}”

Comment#11: Similarly, it might be easier to say “blue water consumption” rather than “net water withdrawals”, for clarity. The water footprint addresses water consumption (evaporation or incorporation into a product), not withdrawals. At line 170, likewise, it might be more appropriate to change “withdrawals” to “consumption”.

Response: We agree with the reviewer and now replace “withdrawals” to “consumption” and “net water withdrawals” with “blue water consumption” throughout.

Comment#12: Lines 193-195: this statement that “The higher nutritional quality of most nations’ diets comes at the cost of higher damage to the environment...” seems rather contentious and a little too bold based on the actual results depicted in this study. Can the authors temper this a bit based on the actual study findings?

Response: We now temper this down and at line 218 simply say that “most nations with high nutritional quality also have high environmental footprints.”

Comment#13: Lines 309-312: the statement about small differences in blue water requirements of plant- and animal-based foods is surprising and counterintuitive. The

authors should carefully check this. It is also contradicted by the result described at lines 385-388 about a sharp drop in freshwater consumption under the VGN scenario.

Response: Note that the result described at lines 340-341 about a sharp drop in freshwater consumption under the VGN scenario is relative to the “current (reference)” scenario. What we meant was that the water footprint under the three alternative scenarios (HGD, VGT and VGN) is almost the same and the fact that while the carbon footprint of plant and animal-based foods differs a lot, the differences in blue water requirements of plant- and animal-based foods is “relatively” smaller than the difference in their carbon footprint.

We now rephrase this sentence at lines 337-347 to be more clear:

“We found that compared to the current (reference) scenario, the blue water footprint decreased under the HGD, VGT and VGN scenarios (except for countries whose caloric intake had to be scaled up to 2300 kcal levels). However, Unlike GHG emissions which progressively decreased substantially from HGD to VGN, the national water footprints under the three HGD, VGT and VGN scenarios are nearly identical/very close/similar. For example, the current blue water footprint (reference scenario) of North America is 348 liters capita-1day-1 while under the HGD, VGT and VGN scenarios it is 227, 214 and 210 liters capita-1day-1 respectively (i.e. a reduction of 35%, 38% and 40% respectively). This is because the differences between blue water requirements for animal and plant based food items are much smaller than the differences in their carbon footprints (e.g. carbon footprint of bovine meat is 50 times higher than wheat but its blue water footprint is just two times higher).”

Comment#14: Line 349: very small nuance, but the authors might say “anthropogenic land used for agriculture” instead of ‘land to produce food’ in terms of land clearing and its biodiversity impacts. Much of the recent land expansion in the tropics has been linked to commodity crops, such as soybean, which are indirectly consumed in foods after being fed to livestock. This is one of the cruxes of the challenge around inefficiencies in our food systems.

Response: We agree and now replace “anthropogenic land use to produce food” with “anthropogenic land used for agriculture”.

Comment#15: Lines 365-370: the current discussion of gender equity and per capita incomes seems rather simplistic. One of many potential references that could help address the issue of gender equity and social justice with respect to food systems and economic development is Schipanski et al. (2016), or a reference on gender equity and social justice therein. --Schipanski, M.E., MacDonald, G.K., Rosenzweig, S., Chappell, M.J., Bennett, E.M., Kerr, R.B., Blesh, J., Crews, T., Drinkwater, L., Lundgren, J.G. and Schnarr, C., 2016. Realizing resilient food systems. *BioScience*, 66(7), pp.600-610.

Response: We agree and have added the following sentence at lines 407-410, which cites the above reference.

“As pointed out elsewhere⁴⁸, the resilience and performance of food systems can be efficiently enhanced by focusing interventions on vulnerable populations, which would be

reflected in improvements of the gender equity and income inequality indicators.”

Comment#16: Lines 393-394: this represents an increase in per capita GHG emissions from currently low levels in a relative sense.

Response: We agree with the reviewer and to avoid confusion, now report the absolute increase in the footprints instead of % increase from currently low levels. The new sentence at lines 433-436 now reads as:

“For example, while adoption of HGD in South Asia has the potential to increase Population Share with Adequate Nutrients (PAN) score by an additional 17%, it also entails an increase in per capita diet related GHG emissions and water use by 0.45 kg CO₂eq. capita⁻¹ day⁻¹ and 69 liters capita⁻¹ day⁻¹ respectively (Table 2).”

Comment#17: Lines 588-591: Can the authors be more specific about the quantity of eggs/dairy ‘allowed’ in the VGT scenario, or at least how these food items are handled?

Response: There is no constraint on eggs/dairy in the VGT scenario. We followed Springmann et al. (2016) guidelines on designing the HGD, VGT and VGN scenarios from the current national diets. According to them, the five constraints applied in the VGT scenario (per day) are - fewer than 50 g of sugar; no red meat or poultry or fish; 2,300 kcal of total energy intake; at least six portions of fruits and vegetables (6 x 80 g), and at least one serving (80 g) of legumes/pulse.

We now mention this and added following sentences in the Methods section lines 694-697 to further clarify this:

“Note that there is no constraint or allowable level defined for eggs/dairy intake in the VGT scenario and it is treated in the same way as other food items such as wheat or rice. Thus, if the current caloric intake of a country is 2300 kcal capita⁻¹ day⁻¹, then the amount of egg or dairy intake levels remain unchanged from reference to VGT scenario.”

Comment#18: Table 1: please consider changing the label for Shannon Diversity to “Shannon Diversity of Consumption”, or something indicating diversity of food supply, to more clearly separate from Food Production Diversity.

Response: We agree and now change the label for Shannon Diversity to “Shannon Diversity of Food Supply” in Table 1 and main text throughout.

Reviewer #2 (Remarks to the Author):

Comment#19: The authors present an ambitious effort to characterize the sustainability of the global food system. The manuscripts builds on several related efforts that proposed either proposed indicators or presented related results. In addition, the authors present a few new indicators. It's clear that a diverse set of expertise worked on this paper.

Although there is a wealth of information, the main messages are not very clear. Very little is done to integrate the findings into a few clear messages. Do the results mean

anything beyond presenting the current status and showing that there's lots of variation around the world? What's new here—does the main message differ from other reports or is the main progress that the indicators are quantified? Were there any surprises?

Response: The novelty of this study can be viewed through three main aspects. First, certainly, the main progress is that through a new framework, we were able to for the first time quantify and compare the 25 sustainability indicator scores for 156 countries across seven domains. The holistic nature of the analysis and bringing all indicators together at one place is one of the biggest strength of the paper (as also mentioned by reviewer#5). Previous global scale analyses have mainly focused on 1-2 or a limited number of indicators. This research gap is mentioned upfront in our abstract:

“The wide scope of the SDGs call for holistic approaches that integrate previously “siloeed” food sustainability assessments. Here we present a first global scale analysis quantifying the status of national food system performance of 156 countries, employing 25 sustainability indicators across seven domains”

By doing so, we found that (abstract lines 9-10):

“different countries have widely varying patterns of performance with unique priorities for improvement ...

Second, the study presents several results and novel findings that are directly relevant for national policy makers. For example, the detailed results per country (provided in Supplementary Tables) can help identify the areas/indicators that need attention as well as position of a country vis-à-vis others globally (abstract lines 11-12):

“Our nation-specific quantitative results can help policy-makers to set improvement targets on specific areas”

For example, at discussion section lines 359-368 while comparing the new nutrition results, we say:

“For example, the average intake of four health sensitive nutrients: sugar, saturated fat, total fat and cholesterol is currently 2, 2, 2 and 7 times higher for high-income nations than the low-income nations.²³ The % population adequacy of Vitamin D, Vitamin E, Calcium and Folate is low even in high income nations (Supplementary Table 3). Such information is valuable in designing different interventions, such as increased production of particular crops⁴⁹, dietary diversification³³, food imports⁵⁰, biofortification⁵¹, tax incentives⁵² etc.”

One of the main findings of this study (as also identified by the reviewers 1, 3 and 4) is that health/nutrition and environment do not always go hand in hand. We now added this point in the discussion section lines 436-437 to further emphasize this:

“Such patterns suggest that diets higher in nutrition are not necessarily more environmentally beneficial.”

Another novel finding is regarding the potential need for nutrient supplementation in nations shifting to plant-based foods (as also mentioned by reviewer#1 above) is at lines 444-447:

“Such patterns suggest that dietary guidelines on less meat (e.g. VGT, VGN) should potentially be accompanied by additional recommendations on the intake of special plant-based foods rich in particular micronutrients such as Brazil nuts for Selenium⁵⁶ or dried purple laver for Vitamin B₁₂.⁵⁷”

Third and finally, apart from presenting the results, the manuscript identifies several existing data gaps and possible future analysis that can build upon our findings to reduce the uncertainty in results and provide additional insights. For example at lines 479-485 we mention:

“Future studies should extend our work by quantifying the impact of food systems through additional indicators (e.g. nitrogen and phosphorus emissions causing eutrophication and acidification) and comparing them to their planetary boundaries (sustainable levels).¹ Combining our metric-based methodology with future scenarios generated by integrated assessment models could provide insights into the impact of various climate change scenarios, socio-economic development pathways, demographic and diet trends, and alternative global trade policies on future global food systems.”

At lines 466-468:

“Next, we used nutrient values of different food items from a single food composition database²² for all countries, although the nutrient composition of similar food items may vary geographically. As detailed national food intake data and localized food composition tables become available, such uncertainties can be reduced.”

Or at lines 570-573:

“Future studies could employ data from new sources, such as the global dietary database (<http://www.globaldietarydatabase.org/>), which are currently under development and aim to provide more refined national dietary intakes of food and nutrients for children and adults by age, sex, pregnancy/nursing status, rural vs. urban residence, and education levels.”

We tried our best to highlight the novelty and main messages of the study while staying within the word limits of the journal. We now made a number of changes in the discussion section upon suggestions by the five reviewers. For example, we now deleted several sentences (e.g. the correlation of environmental footprints with GDP or the explanation on high biodiversity footprint of different nations) with details that were redundant and already presented in the results section and focus more on key messages.

Comment#20: The text needs a lot of editing to have the main messages be distinguished from the details. It currently reads as a group edited, indicator-by-indicator prose in the methods, results and discussion sections. It should be a bit more formal as well, changing phrases that are informal such as “score well on most fronts” and “stood close to”.

It was challenging for me to comment on all aspects of the manuscript as it covers (and requires) such a broad set of expertise. However, for all indicators, the authors cited references that demonstrated that they used accepted methods to compute the metrics. These are summarized in Table 1 as well as throughout the text. They also acknowledged shortcomings of the approaches and data limitations.

Response: We now removed the informal phrases upon suggestion by the reviewer and also made a number of edits in the manuscript in order to better highlight the main messages and incorporate the suggestions made by the five reviewers. Please see the response to comments and main text for all edits made.

More specific comments:

Comment#21: Given the uncertainty in the values for the computed metrics, can you really use the results to “set quantitative goals (e.g. reducing per capita GHG emissions below the threshold level of 2.055 kg CO₂ eq.)”? Elsewhere the authors state that “We found that the carbon footprint values per country have high uncertainty, varying by over an order of magnitude.” Similar uncertainties will arise in estimates of food production in some countries if relying on FAO data. As a result, there will be high uncertainty in all the nutrition metrics.

Further, while aggregating indicators to a 0-100 scale makes it easier to compare across aspects of the food system, but it also indirectly assumes that all the indicators are equally weighted and the factors (environmental, nutrition, etc.) are independent. Obviously, neither assumption is true. Are these main results or national level metrics better used to identify priorities and track directional progress rather than setting specific goals?

I agree that the authors approach is the best that can be taken given the data limitations. I just don’t think the utility should be oversold to the point of setting or measuring progress toward quantified goals with any high level of precision.

Response: We agree with the reviewer that given the high uncertainties in the underlying data, the current results might not be ready to inform quantitative goal setting. We therefore remove the claim that our results can be used for quantitative goal setting from the concluding paragraph of the main text and just say that the results and approach can be used to identify priorities and track progress. The rephrased concluding paragraph at lines 503-508 now reads as:

“Our nation-specific quantitative results can be used by national and global policy-makers to identify key areas of improvement and set priorities. The multi-indicator approach enables one to evaluate the impact of alternative strategies aimed at a particular aspect (e.g. reducing carbon footprint by replacing ruminant meat with dairy products or pulses) while at the same monitoring the impact on other food system performance metrics (e.g. Food Nutrient Adequacy).”

However, we still think that mentioning the carbon footprint vis-à-vis some form of threshold is useful to see the results in a context. Instead of saying that many countries exceed the threshold, we now say that the calculated “mean” carbon footprint exceeds

the threshold and that high uncertainty renders this comparison statistically insignificant. We now explicitly acknowledge this uncertainty in results at lines 164-168:

“Currently 81 of 156 countries (representing ~52% of global population) have mean dietary carbon footprints above this threshold (Fig. 2a). However, this comparison is not statistically significant due to high uncertainty in the emission factors of food products²⁵. The calculated national carbon footprints vary over an order of magnitude around this threshold (Supplementary Table S1).”

Comment#22: There is little comparison of the results with previous studies. Although other studies have not addressed the full set of metrics as the work here, there has been substantial work on several of the specific issues. These include malnutrition/stunting and GDP, diets and GHGs, water and diets, etc.

Response: We now include following sentence in the discussion section lines 448-457 regarding comparison with previous studies:

“Overall, our results and trends in the GHG and water use consequences of dietary change are consistent with previous studies who also found that reducing animal sourced food leads to reduction in national water^{8,14,60} and carbon footprints.^{10, 11, 15, 55} Differences in absolute values of the footprint between ours and previous studies might occur due to a number of factors such as different year of the analysis, use of different dietary scenarios, different food product emission factors²⁵, inclusion of land use change in product carbon emission factors, etc. We are not aware of studies quantifying nutritional quality of global diets considering multiple nutrients and indicators such as done by us, although it is well known that high-income nations have higher intake of nutrients of health concern (e.g. sugar, fats) in contrast with low-income nations where less than recommended intake of essential nutrients is more prevalent.^{4,12,33}”

Comment#23: Table 2 is difficult to interpret without the current values. I suggest that the current diet/consumption numbers are added to the chart. That would put the change values into context. However, it still does not address the problem of the metric values being intuitive as have no idea if a change of 5 is a lot or little. Is there a better way to present the changes so that they are more intuitive?

Further, it's helpful to have the regional summaries here, similar to Figure 4. But how were the regional values calculated? Are they per capita, average of country values, etc.?

Response: Thanks for this suggestion, we agree with the reviewer and have now added another row in the table (labeled as REF) showing the indicator values under current diets. In addition, we added the following sentences in the footnote to the Table 2 at lines 326-331 in order to put the values into context and explain how the regional values were calculated:

“The region-aggregated indicator scores were calculated by taking the average of all national indicator scores within the region. To put the values into context, a NBS and DNS value of 100 implies perfectly nutritious diets. A PAN score of 100 implies that 100% of the region's population is meeting daily nutritional requirements (see methods for details). The

current global average carbon and water footprint is 2.3 kg CO₂eq. capita⁻¹ day⁻¹ and 237 liters capita⁻¹ day⁻¹ respectively.”

Comment#24: Title: I don't have any immediate suggestions for changes, but I generally don't think of nutrition and food safety when I think of sustainability. Also, why the plural for systems? Is it because the analysis is at the country scale or because of the scenarios?

Response: The plural for systems is because the analysis is for 156 countries. We decided not to change the title because the paper deals with 25 sustainability indicators of food systems and sustainability is a broad term encompassing economic, environmental and social aspects. Nutrition and food safety fall within the social aspects.

Comment#25: The scope of the paper is very broad, but should address a few key items in the discussion. For example, climate change is only referenced in terms of GHG emissions and resilience. How will a warming world affect the indicator values? Do the trends in population and diet add constraints? Over a third of all calories are traded—could many of the problems in the global food system be fixed by trade?

Response: Thanks for the suggestions, which we have now used as the basis for an additional statement at lines 481-485 in the discussion section:

“Combining our metric-based methodology with future scenarios generated by integrated assessment models could provide insights into the impact of various climate change scenarios, socio-economic development pathways, demographic and diet trends, and alternative global trade policies on future global food systems.¹⁸”

Comment#26: Resilience - crop diversity may be a good indicator, but other factors are likely more important in most places. These include: irrigation, nutrient management, soil organic matter, and crop selection.

Response: As noted above, the primary intent of this paper is not to defend each indicator and metric, but rather to simply calculate current country-level values using a reasonably comprehensive set of food system performance measures. Having said that, the other indicator of Resilience (ND-GAIN Index) is calculated based on 45 different factors that affect the resilience of a country's food systems and it does include some of the factors highlighted by the reviewer. We now shortly mention this at lines 254-255:

“.....and is calculated based on the status of 45 different factors (e.g. flood hazard)”

Comment#27: Is one of the main solutions to reduce income and gender inequality? Are there places where gender and income equality measure are high but not adequate food and nutrition? Related, the income and gender inequality metrics provide insights into sub-national variations. Would your main results be different if other indicators were sub-national? Or do you think the main patterns would be the same?

Response: Published research and input from stakeholders on a previous paper (Gustafson et al. 2016) have both highlighted the importance of including such inequality indicators in holistic measures of food system performance. Yes, these

inequality measures are dependent on the degree of sub-national variation, but the actual spatial scale is still national and it was beyond the scope of this paper to delve further deep into this discussion. Please see our response to the comment#4 above where we acknowledge that a sub-national analysis can provide better insights into the status of several indicators.

Comment#28: Diversity indices such as the Shannon index are nice for aggregating information. But the metric is very abstract and there are different pathways for getting similar index values. How does this limit the indicators utility for guiding action?

-Paul West

Response: We repeat that the purpose of this paper was not to defend individual indicators, and by including multiple complementary indicators, we strived to compensate for the limitations of individual indicators. We respond here that the Shannon Index is used across multiple scientific disciplines and is perhaps the simplest and most popular measure of diversity in the overall scientific literature. In this particular context, MFAD is probably a somewhat more relevant measure of diversity – albeit more complicated. However, by including both of them as 2 of the total of 6 indicators of Food Nutrient Adequacy, we believe the right balance has been struck.

Reviewer #3 (Remarks to the Author):

Comment#29: This is an excellent and important paper. The data is well-presented, with appropriate caution. The results are of great policy significance. The style is clear and readable. I strongly support publication of this paper with no significant amendments. The only caution is that I am not a professional nutritionist so their critical review comments may be more important.

They present the first global scale analysis quantifying the status of national food system performance of 156 countries, employing 25 sustainability indicators across seven domains. Their nation-specific quantitative results will help policy-makers to set improvement targets on specific areas to improve nutritional intake and sustainability.

Our biggest health and nutrition challenge is to provide a balanced diet to a growing world population in a sustainable manner. Their seven metrics of sustainable nutrition security are (1) Food Nutrient Adequacy; (2) Ecosystem Stability; (3) Food Affordability and Availability; (4) Sociocultural Wellbeing; (5) Food Safety; (6) Resilience; and (7) Waste and Loss Reduction (see Table 1). Each of the metrics comprises multiple indicators that are combined to derive an overall score (0–100). They quantify the food system sustainability status of 156 countries for the year 2011 using the seven metrics and 23 underlying with two additional indicators on biodiversity impacts and health sensitive nutrient intake. They also calculate a new indicator, the Disqualifying Nutrient Score (DNS), by comparing the total daily intake of five public health sensitive nutrients (sugar, sodium, cholesterol, saturated fat and total fat) with their Maximal Reference Values (MRV). Countries with lower intake of these sensitive nutrients score higher on this indicator. They also calculate the Population share with Adequate Nutrients (PAN),

the Ecosystem Stability metric, and the land and biodiversity footprint of each country's daily food consumption.

Finally they constructed three alternative dietary scenarios [healthy global diets (HGD), lacto-ovo vegetarian (VGT), and vegan (VGN)] based on global dietary guidelines on healthy eating and recommended levels of fruits, vegetables, red meat, sugar and total caloric intake for each of the 156 countries.

I'm a paediatrician working in policy and practice but not a professional nutritionist. The authors present a good narrative in the discussion about uncertainties and limitations in interpreting the results which seems clear and reasonable. For example, the FBS provides information on only 94 food items. They used nutrient values of different food items from a single food composition database, and report that carbon footprint values per country have high uncertainty.

They present good ideas for future studies. For example, quantifying the impact of food systems through additional indicators (e.g. nitrogen and phosphorus emissions causing eutrophication and acidification), and developing more sophisticated scenarios with constraints on other food groups, considering consumer preferences, with additional nutritional, environmental, economic, and social indicators of sustainability, are important areas for future research.

Key results that caught my eye as highly relevant for policy:

1. *Income*: A positive effect of higher income on national nutritional diversity and adequacy (Table 1). However, there was a strong negative correlation between DNS and GDP.

2. *Ecosystem Stability*: Carbon footprint varies from around 0.7 kg CO₂eq capita⁻¹ day⁻¹ for African countries such as Mozambique, Ethiopia and Malawi to >4 kg CO₂ eq. capita⁻¹ day⁻¹ for New Zealand, Australia, USA, France, Austria, Argentina and Brazil . Per capita food related GHG emissions with "sustainable levels," at no more than 50% of the yearly total per capita emissions shows 81 of 156 countries (representing ~52% of global population) have GHG emissions above this threshold. Bovine meat consumption contributes the most to the GHG emissions. "Blue water footprint" is similar to carbon footprints.

3. *Women's empowerment*: A very weak correlation with GDP in Gender Equity which directs policymakers to pay more attention to, and invest more, in women's empowerment. Gender equity remains a big problem, suggesting that women's empowerment concerns persist even as national incomes rise.

4. *Waste and Loss Reduction*: Higher-income countries generally waste much larger percentages of their food at the post-consumer level than do lower-income countries. However, pre-consumer losses in lower-income countries are relatively large. They measured the produced food that is not either lost (pre-consumer) or wasted (post-consumer) in a country. High-income nations such as Canada, USA, Australia and EU countries score lower (~60) than low or lower-middle income nations

5. *Alternative dietary scenario outcomes*: Dietary changes toward fewer animal and more plant-based foods result in significant reductions in daily per capita disqualifying

nutrient intake (e.g. >50 point increase in DNS for North America and Europe) but relatively small improvements in the Nutrient Balance Score. They found that adopting diets low in animal sourced foods can significantly reduce the food related per capita GHG emissions and freshwater use of nations (e.g. >70% and 40% reductions, Adoption of alternative diets (healthy global diets (HGD), lacto-ovo vegetarian (VGT), and vegan (VGN)) leads to high improvement in Disqualifying Nutrient Score (DNS) across Europe, Latin America and North America. Also they would result in food related per capita GHG emissions 12, 40 and 50% lower than under current diets respectively, and where at present 81 out of 156 countries exceed the sustainable threshold level (2.055 kg CO₂ eq. capita⁻¹ day⁻¹), the number of countries not meeting this criterion drops to 73, 8 and just 7 under the HGD, VGT and VGN, respectively.

6. *Diet in the wealthy west:* Dietary changes in North America and Europe lead to the largest reductions in per capita carbon footprint primarily because of a decrease in total caloric intake and reduced meat intake levels under the three scenarios. The average intake of five health sensitive nutrients: sugar, sodium, saturated fat, total fat and cholesterol is currently 2, 5, 2, 2 and 7 times higher for high-income nations than the low-income nations. Policymakers can focus on increased production of particular crops, dietary diversification, cut food imports, biofortification, tax incentives and so on.

7. *Food and biodiversity:* The current rate of species extinction is over 100 times the background rate, so anthropogenic land use to produce food is one of the major drivers of global biodiversity loss, and the land footprint is not a good proxy for the biodiversity footprint.

Response: We thank the reviewer for encouraging words and finding our results useful.

Reviewer #4 (Remarks to the Author):

Comment#30: The paper proposes an analysis of the sustainability of national food systems of 156 countries using 25 sustainability indicators defined by Gustavson et al. and available global data. It is a useful enterprise, that confirms more partial analysis. It also brings forward some interesting points, like the fact that the higher nutritional quality of diets can go against to the environment (193-195), a fact that is often not considered in global papers that insist on environment and health go hand in hand. This point would deserve to be better emphasized, including in the discussion.

Response: We agree with the reviewer that one of the main results of this study is that health/nutrition and environment do not always go hand in hand. We now added this point in the discussion section lines 436-437 to emphasize this: *“Such patterns suggest that diets higher in nutrition are not necessarily more environmentally beneficial.”*

The relevant paragraph in the discussion section lines 431-437 now reads as : *“...we also find that the above dietary change scenarios do not always lead to twin nutrition and environmental benefits for all countries through all indicators. For example, while adoption of HGD in South Asia has the potential to increase Population Share with Adequate Nutrients (PAN) score by an additional 17%, it also entails an increase in per capita diet related GHG emissions and water use by 0.45 kg CO₂eq. capita⁻¹ day⁻¹ and 69*

liters capita⁻¹ day⁻¹ respectively (Table 2). Such patterns suggest that diets higher in nutrition are not necessarily more environmentally beneficial.”

Comment#31: Given that this paper is likely to be of interest for a very diverse set of researchers it would benefit from being clearer on some points, in particular on the way some indicators are calculated or applied. For instance, it seems that the indicators are calculated/applied to the consuming country; it needs to be said. It would also be good to recall, line 33-34 that consumption and production spaces are now quite different, especially for some products that are widely traded. In other words some of the impacts determined by consumption are different depending on the place/country of production.

Response: We agree with the reviewer and now explicitly mention this in the Methods section lines 667-668:

“Note that all results are based on national consumption rather than national production (i.e. taking into account of where the foods were produced and attributing the impacts to the importing country).”

Comment#32: Some of the indicators, like the environmental performance index are not determined only by food systems (mainly but not only), it should be said. For other, in particular gender equity it is not clear if its scope is the food system or the whole country. This is important given the prominence of women labor at most stages of the food value chain.

Response: The reviewer is correct that some indicators are determined not only by food systems. Where possible, we considered the indicators impacted by national food systems alone. However, a lack of data did not allow us to do so in case of six out of 25 indicators selected in this study. We now mention the indicators that are exception to this at method section lines 518-523:

“Note that some of these national indicator scores are determined not only by food systems but other sectors/industries as well (e.g. mining). For this study, we specifically considered the impacts of national food systems alone where the available data allowed us to do so (e.g. carbon, water, land footprints associated with national food consumption alone). However, a lack of data did not allow us to do so in case of six out of 25 indicators selected in this study. These exceptions include: non-renewable energy use³⁵, poverty index³⁶, income equality³⁷, gender equity³⁸, respect for community rights⁴⁰, and ND-GAIN index⁴² (see ref. 18 for full details on each indicator.”

For the environmental performance index (EPI), we added following sentences in the methods section lines 648-652 to highlight this:

“We used the ‘environmental performance index (EPI)’ as a proxy for food system indicator of ‘Ecosystem Status’.¹⁸ The EPI ranks the performance of different countries in two broad policy areas (ecosystem protection and human health risk from environmental harm) through a suite of indicators.³⁴ We took a simple average of the following five indicators of ecosystem protection (each already available at 0–100 scale) relevant to food systems to

calculate ecosystem status indicator for this study: Agriculture, Fisheries, Water Resources, Forests, and Biodiversity/Habitat.”

Comment#33: Precisely because some indicators seem to be country wide and not only national food system wide, it would be good to be precise for instance at line 140 that it is the carbon foot print of the diet. Same for other indicators for which we are used to see per capita economy wide values like the water footprint.

Response: We agree with the reviewer now mention that these are dietary footprints associated with per capita daily food consumption at various points in the manuscript.

Comment#34: For some indicators, like for food and losses for instance, the data used has not been constructed to be downscaled at national level and thus loses certainty when downscaled (for a short discussion on the methodology used and its accuracy see HLPE (2014) *Food losses and waste in the context of sustainable food systems*. A report by the High Level Panel of Experts on Food Security and Nutrition of the Committee on World Food Security. Rome: FAO. Box 1 p28.

Response: We now acknowledge this source of uncertainty in the methods section lines 523-525:

“For some indicators, such as food waste and losses, the data is available for certain aggregated world regions only and needed to be downscaled to national level for this study, thus potentially compromising its accuracy.”

Comment#35: The biggest issue however concerns the economic and social dimensions. There is no indicator of the economic (and social) sustainability of the food system itself. A point which was already noted in Gustafson et al, but without taking the full measure of its consequences. In fact, with the main economic indicator being affordability it seems that sustainability would be the cheapest food possible which does not account neither for the fact that most hungry and malnourished are actually food producers, and that there could be no long term sustainability to the sector without enough investment, returns on investment nor attractiveness for young workers without decent income. Even if it may not be easy to define such an indicator (of economic health of the system), it is a major gap that absolutely needs to be highlighted.

Response: This is a very reasonable critique of the previously published paper (Gustafson et al. 2016), upon which we have fully relied in order to conduct this analysis. Adding new indicators to cover these kinds of social and economic effects is beyond the scope of this paper. However, in light of the strength of this comment, we have added the following statement to the beginning of the discussion of our socio-economic results at lines 396-400:

“Before discussing the socio-economic results, we acknowledge an important limitation in the underlying indicators upon which we rely: namely, the lack of any measures of the overall economic health of the various players within the food system itself: i.e. producers, food transport and processing, retail, food service industry, etc. – all being highly relevant except in the special case of subsistence farming.”

Comment#36: The discussion on the alternative scenario outcomes should also give room to potential outcomes of substitutions, for environmental impacts (potential increase of GHG emissions due to greenhouses in some colder countries when vegetables would substitute ASF (see for instance Tom M S, Fischbeck P S & Hendrickson CT (2015) *Energy use, blue water footprint, and greenhouse gas emissions for current food consumption patterns and dietary recommendations in the US*. *Env Syst* Dec 1–12.)

And importantly, the lack of proper indicator on the economic and social sustainability of food systems does not enable even to consider potential impacts on farmers, and especially on small farmers of a reduction of ASF consumption, livestock representing in many countries a critical way to increase economic productivity of both land a labor and to escape poverty.

Response: We agree with the reviewer and have now added the suggested reference to Tom et al. as well as following sentences in the discussion section lines 494-499:

“We could not evaluate the full implications that such dietary transformations would have on the environment, social and economic aspects. For example, increasing the intake of vegetables/fruits and replacing animal sourced foods with vegetarian substitutes might adversely affect the income of small livestock farmers in some regions. In certain cases, such substitutions can also increase the carbon footprint in colder countries due to an increase in energy use for producing vegetables in greenhouses (Tom et al. 2016).”

More detailed comments:

Comment#37: 25-26, no mention at all of economic issues. Food systems are by far the biggest employer in the world, particularly in poor countries, and they also represent an important part of GDP, not only in developing countries, but also in developed countries with a different distribution of income generated between agriculture/transformation/distribution.

Response: We agree with the reviewer and add above points suggested by the reviewer as well as the fact that improvement of economic security is an additional challenge that food systems must tackle. The new sentence at lines 27-34 now reads as:

“Food systems are by far the biggest employer in the world, particularly in economically poor countries. They also represent an important part of national GDP although the share of income generated from agriculture production, food processing, sales and distribution varies across different countries. Going forward, global food systems need to ensure improved economic security of actors involved and combat existing malnutrition/obesity related health problems while keeping the environmental impacts low enough so as not to transgress the planetary boundaries of biophysical processes and further destabilize earth systems.”

Comment#38: 182-183 extensive grazing is also linked to very specific ecosystems and species, that are in fact preserved by this mode of land use. It should be mentioned here. As rightly said later (350) the land footprint is not a good proxy for the biodiversity footprint.

Response: We agree with the reviewer and have now mentioned this at lines 214-216:

“Figure 2 shows that nations with higher land footprint do not necessarily have higher biodiversity footprints (e.g. Mongolia) because certain land uses such as extensive livestock grazing are less harmful to species than intensive cropland.”

Comment#39: 223, as explained above, you cannot say that the previous indicators dealt with economic considerations, rather economic access for consumers.

Response: Thanks for the suggestion; we now replaced ‘economic considerations’ with ‘economic access for consumers’.

Comment#40: 269-270, propose to write rather "of disqualifying nutrients" and to delete sugar from the parenthesis as the link between meat consumption and sugar intake seems dubious.

Response: Done.

Comment#41: 380 add "improving before "food safety".

Response: Done.

Comment#42: Methods: you may want to complete this section as the method is missing for some of the indicators, or at least give an easily reference to be consulted.

Response: We agree with the reviewer. Although the sources of data for all indicators were listed upfront in Table 1 of the main text, we now explicitly mention this in Methods section lines 515-518 and direct the reader to Table 1:

“Data for the indicators associated with metrics 3-7 (Food Affordability & Availability; Sociocultural Wellbeing; Food Safety; Resilience; and Waste & Loss Reduction) per country were directly imported from various sources and converted to a 0-100 scale using Eq. 5 below. Please refer to Table 1 for sources of data for all indicators.”

In addition, we now added details on calculation of several indicators. Please see our response to comment#32 and #34 above.

Reviewer #5 (Remarks to the Author):

Comment#43: This is a thorough report of an analysis of the sustainability of food systems in 156 countries, which develops metrics across multiple indicators to assess the current nutritional status and environmental impact of food production and consumption. The authors should be commended for their hard work bringing multiple metrics together in a single place. The holistic nature of the assessment of the impact of the food system is the biggest strength of this paper.

Response: We are glad that the reviewer found the paper holistic and thorough.

Major comments:

Comment#44: Page 4, lines 74-76: “The national daily average intake of each nutrient was obtained by combining a food composition database with the FAO food balance sheet (FBS) database.” What about nutrients added during processing? This approach

will give very little information about sodium levels in food. Indeed, Table S2 suggests that sodium levels have been greatly underestimated by their approach. The region with the highest level of sodium consumption is North America (1216mg capita⁻¹ day⁻¹), but this level is far lower than the MRV of 2400mg (equivalent to 6g of salt). But mean consumption of salt in the UK (for example), estimated by 24 hour urine collection (the gold standard), is about 8 g day⁻¹ (estimates from the 2014 National Diet and Nutrition Survey) - far higher than levels estimated for this analysis. I suggest that the authors should remove sodium from their list of disqualifying nutrients.

Response: We thank the reviewer for pointing this out. We agree that our current values do not take into account sodium (salt) added during cooking or processing steps and therefore might underestimate the amount of sodium intake in each country. We therefore now remove sodium from the list of disqualifying nutrients and recalculate the national disqualifying nutrient scores. We also compare the sodium intake value estimated from urine samples in National Diet and Nutrition Survey with those estimated from FAO food balance sheet (FBS).

This is now mentioned in methods section lines 611-612 shortly:

“Unlike Fern et al.²⁰, we did not include trans fatty acid and sodium for DNS calculations (see supplementary information for details).”

Owing to word limit constraint in methods section, we now mention the details on removing sodium in Additional Methods in supplementary information:

“We did not include ‘sodium’ as a disqualifying nutrient since it is often added during processing/cooking in the form of salt. The intake amounts of sodium calculated from FAO’s food balance sheets (FBS)²³ might thus be underestimates. For example, the mean sodium intake in the UK estimated from FBS²³ is around 1.2 g capita⁻¹ day⁻¹ which is around three times less than the mean value of 3.2 g capita⁻¹ day⁻¹ (equivalent to 8 g capita⁻¹ day⁻¹ of salt) estimated from 24 hour urine samples in the 2014 National Diet and Nutrition Survey (<https://www.gov.uk/government/statistics/national-diet-and-nutrition-survey-assessment-of-dietary-sodium-in-adults-in-england-2014>).”

Comment#45: The authors do not mention the Global Dietary Database (GDD), which combines FBS estimates with population surveys of dietary intake in order to model comparable estimates of food and nutrient intake at a country level. The GDD approach overcomes many of the limitations of using FBS estimates and this analysis would have benefited from using data from the GDD (which may not have been possible). At the very least, the authors should comment on the GDD when discussing the limitations of using FBS data for characterizing food and nutrient intake.

Response: We thank the reviewer for this suggestion and now mention this in the Methods section when discussing the limitations of using FBS data at lines 570-573:

“Future studies could employ data from new sources such as global dietary database (<http://www.globaldietarydatabase.org/>) that are currently under development and aim to provide more refined national dietary intakes of food and nutrients for children and

adults by age, sex, pregnancy/nursing status, rural vs. urban residence, and education levels.”

Comment#46: What is the justification for the choice of nutrients for both the NBS and the DNS? I am surprised to see both cholesterol and total fat included in the DNS.

Response: We shortly mention this in method section lines 610-611:

“For this study, we simply followed Fern et al. in selecting the 25 qualifying nutrients for NB score and four disqualifying nutrients for DNS, although others have used anywhere from 5-23 nutrients.”

Owing to word limit constraint in the methods section, we now added following paragraph in the Additional Methods in supplementary information to discuss our choice of nutrients which is based on study by Fern et al. (2014):

“One may argue our selection of a large number of nutrients to carry out the nutrition balance (NB) and disqualifying nutrient score (DNS) calculations owing to potential redundancies (e.g. including both cholesterol and total fat in DNS). Choice and selection of nutrients also vary across national guidelines. This is acknowledged by Fern et al.²⁰, who mentioned that past studies^{45,64} have used anywhere from 5-23 nutrients while calculating the nutrient density of foods (providing almost similar results) and the NB, DNS calculations can also be based on a smaller number of nutrients. Clearly, globally harmonized and consensus-based set of guidelines are needed on selection of particular nutrients to calculate the nutritional quality assessment of diets. For this study, we simply followed Fern et al.²⁰ in selecting the 25 qualifying nutrients for NB score and four disqualifying nutrients for DNS. The main criteria in selecting these particular nutrients was whether their daily required intake (DRI) values have been published by the Institute of Medicine, National Academy of Sciences and whether the nutrient compositional data for them existed in current databases such as USDA SR.²² The detailed justification for the choice of these nutrients can be found in Fern et al.²⁰.”

Comment#47: Estimating the blue water impact of each country: was this done with the region-specific results from Mekonnen & Hoekstra? If so, how did the authors gain country-level estimates from the sub-country regional estimates reported in Mekonnen & Hoekstra?

Response: We did not use the water footprint values derived by Mekonnen & Hoekstra (2010) at the grid level (5 by 5 arc minute) and then aggregated to a country level. This is because these values are old and correspond to crop production and their water use between the period 1992-2005.

Instead, we follow the bottom-up approach to calculate the national water footprint that has been applied by many studies in the past (e.g. Vanham et al. 2013). Here, we first import the global average blue water footprint values for different primary crops, derived crop products and animal products (in liters gram⁻¹) provided by Mekonnen & Hoekstra (2010a, 2010b). We then multiply these footprint values with their intake levels (in g capita⁻¹day⁻¹) for the year 2011. This is because country or region specific estimates for each of the 94 FAO aggregated food items is not available.

We now paraphrase this sentence in order to be more clear. The revised sentence at method section lines 640-644 now reads as:

“For calculating the national blue water footprint, we follow the bottom-up approach.⁸ We first import the global average blue water footprints (liters g⁻¹) of 94 FAO food items available from Mekonnen & Hoekstra^{26, 27} and multiply them with their respective intake amounts (in g capita⁻¹ day⁻¹) in each country for the year 2011. Summing these up provided the diet related national average daily blue water footprint (in liters capita⁻¹ day⁻¹).”

Comment#48: Presumably all of the results are based on national consumption rather than national production (i.e. taking account of where foods were produced (and importing environmental footprints to the importing country)? This should be explicitly stated.

Response: Yes, the reviewer is correct. Please see our response to comment#31 above. We now include this statement in the Methods section lines 667-668:

“Note that all results are based on national consumption rather than national production (i.e. taking into account of where the foods were produced and attributing the impacts to the importing country).”

Comment#49: The discussion does not provide much insight. It generally just restates the results and states that these may be important for setting country-specific strategies for addressing the sustainability of food systems. Lines 317-404 could be shortened considerably.

Response: We agree and now deleted several sentences that were restating the results (for example the paragraph restating the correlation between indicators and GDP has now been removed). Instead, we now added several points in the discussion section as suggested by the five reviewers. Please see the revised discussion section in the main text.

Minor comments:

Comment#50: Page 26, equation 2: Please can you give definitions of E_k and $a_{q,k}$.

Response: We now add following lines 588-589 just below equation-2 to define E_k and $a_{q,k}$:

“Here E_k is the total daily caloric intake for country k (in kcal capita⁻¹ day⁻¹) and $a_{q,k}$ is the daily intake amount of a qualifying food nutrient q in country k (in g capita⁻¹ day⁻¹).”

Comment#51: Page 7, lines 119-122: “we found that the global median value was 75.73% (averaged over 17 essential nutrients) meaning a significant share of world population (ca. 25%) is not receiving adequate nutrients currently (Table 1)” This statement only makes sense if the global median is calculated with weighting for different populations between countries. Is that the case? Even if it is, I still don’t think

the 25% figure makes much sense, as the 75.73% figure is a median of means of 17 micronutrients. So if a micronutrient was added for which everyone had sufficient consumption, then all of the country-level means would be adjusted upwards and the global median would be adjusted upwards, which would suggest that a smaller proportion of the global population was not receiving adequate nutrition, even though no change has been made to the underlying data on the 17 micronutrients currently included. In summary, I suggest that the statement that 25% of the global population do not receive adequate nutrition should be dropped!

Response: We agree with the reviewer and have now dropped this statement.

REVIEWERS' COMMENTS:

Reviewer #1 (Remarks to the Author):

The revised manuscript has been improved considerably. The authors have carefully and quite thoughtfully addressed the five expansive sets of reviewer comments. In particular, I think the Discussion section is now much more nuanced and the Methods much clearer. This study will be of interest to a broad array of food systems researchers and those in the policy community.

While I do honestly think that there is more room to present these data in a more captivating and illustrative way without too much work (i.e., simple scatterplots), I suppose this is ultimately a 'moot' point in that the core data used are included in the supplemental information file (thanks to the authors for clarifying this and adding the useful new Table S6); the textual description is quite strong. Likewise, it still seems to me that other relevant indicators not considered in the previous two studies could have been relevant substitutes for this one, but I see the neatness of being consistent in building from those past studies. At the scale and scope of this study, one could continue to debate such relatively minor details. My main concerns with the manuscript have now been addressed and I think the authors have also made considerable strides to address the constructive criticisms of three other reviewers.

One very minor final comment: At lines 345-347, the point about the 'relatively' small change in blue water consumption under the the three alternative scenarios compared to the carbon footprint change is because most of the difference is accounted for in the green water component (which changes by a large amount)? I tried to look into the footprint tables from Mekonnen & Hoekstra and see support for this. So, again very minor, but to avoid similar confusion from other readers, perhaps add literally just a few words about "green water" / "rainwater" to this parenthetical sentence?

Reviewer #2 (Remarks to the Author):

The authors made minor revisions to address the comments or defend their choices. I hear your argument that the intent of the paper is not to defend individual metrics. However, if the indicators are not defended and explained why you chose them over many other options, how are you or the readers confident that they are not redundant? You assume that "including multiple complementary indicators, we strived to compensate for the limitations of individual indicators." Couldn't you just as easily assume that the multiple indicators compound problems with the metric rather than compensate for uncertainties?

Reviewer #4 (Remarks to the Author):

The points raised by the reviewers seem to me to have been very well addressed.

I just have some further minor comments:

I 12-13, I would suggest to replace "protein" by "food", as in some countries protein intake is well above needs.

I 107, 109, 153, 232, 257, 284, 363, 429, I suggest to replace "nation" by country; as the meaning of "nation" could differ depending on countries.

231, add "the" before "proportion"

252, add "the" before "resilience"

308, add "s" at the end of "patter"

378, I would suggest to keep "food consumption" rather than "food systems" as what is dealt with here is only the impact from consumption. There are other means of reducing impacts on the environment in the food system, acting on transport, conservation, transformation, agricultural production...

399-400, I would strongly suggest to delete "except in the special case of subsistence farming". Why wouldn't their economic health be relevant? So called subsistence farming is not immune to economy and, most of the time, also linked to it.

436 and 444, suggest to replace "patterns" by "results"

441 Either delete "the" before "scaling up" or add "of" after it.

453 suggest to add "or not" after "inclusion"

Reviewer #5 (Remarks to the Author):

Thank you for your responses to my comments. In all but one instance your responses have been satisfactory and the manuscript has been improved.

However, I still have a concern regarding the way that bluewater use is estimated for the paper. The authors have included the following in the revised manuscript:

"For calculating the national blue water footprint, we follow the bottom-up approach. We first import the global average blue water footprints (liters g-1) of 94 FAO food items available from Mekonnen & Hoekstra and multiply them with their respective intake amounts (in g capita-1 day-1) in each country for the year 2011. Summing these up provided the diet related national average daily blue water footprint (in liters capita-1 day-1)."

I think the term 'bottom-up' is misleading here. The authors use it as the country-level estimate of bluewater use is the sum of the bluewater use for each of the food commodities consumed in the country. However, it implies that the global estimate of bluewater use is a sum of region-specific bluewater use, and this is not the case (as global estimates of the bluewater use are used at the local level). So I would remove the term 'bottom-up approach'.

Secondly, the authors suggest that they used global estimates of bluewater use as these were the only estimates that were available and justify this by saying that this is common practice in the scientific literature. However, I am aware of some projects that use region-specific bluewater use estimates (e.g. the IMPACT model developed by IFPRI has region-specific estimates in its water model). This is a significant limitation of the study and it should be mentioned in the discussion section.

REVIEWERS' COMMENTS:

Reviewer #1

Comment#1: The revised manuscript has been improved considerably. The authors have carefully and quite thoughtfully addressed the five expansive sets of reviewer comments. In particular, I think the Discussion section is now much more nuanced and the Methods much clearer. This study will be of interest to a broad array of food systems researchers and those in the policy community.

While I do honestly think that there is more room to present these data in a more captivating and illustrative way without too much work (i.e., simple scatterplots), I suppose this is ultimately a 'moot' point in that the core data used are included in the supplemental information file (thanks to the authors for clarifying this and adding the useful new Table S6); the textual description is quite strong.

Likewise, it still seems to me that other relevant indicators not considered in the previous two studies could have been relevant substitutes for this one, but I see the neatness of being consistent in building from those past studies. At the scale and scope of this study, one could continue to debate such relatively minor details. My main concerns with the manuscript have now been addressed and I think the authors have also made considerable strides to address the constructive criticisms of three other reviewers.

One very minor final comment: At lines 345-347, the point about the 'relatively' small change in blue water consumption under the three alternative scenarios compared to the carbon footprint change is because most of the difference is accounted for in the green water component (which changes by a large amount)? I tried to look into the footprint tables from Mekonnen & Hoekstra and see support for this. So, again very minor, but to avoid similar confusion from other readers, perhaps add literally just a few words about "green water" / "rainwater" to this parenthetical sentence?

Response: Upon the reviewer's suggestion, we now include the following sentence about the green water component. The revised sentence at lines 284-288 now reads as:

"This is because most of the difference is accounted for in the green water (rain water) component (not considered here). The differences between blue water requirements for animal and plant based food items are much smaller than the differences in their carbon footprints (e.g. carbon footprint of bovine meat is 50 times higher than wheat but its blue water footprint is just two times higher)."

Reviewer #2

Comment#2: The authors made minor revisions to address the comments or defend their choices. I hear your argument that the intent of the paper is not to defend individual metrics. However, if the indicators are not defended and explained why you chose them over many other options, how are you or the readers confident that they are not redundant? You assume that "including multiple complementary indicators, we strived to compensate for the limitations of individual indicators." Couldn't you just as easily assume that the multiple indicators compound problems with the metric rather than compensate for uncertainties?

Response: We understand the reviewer's concerns on the choice and potential redundancy of the 25 indicators used in this study. This point was also raised by reviewer#1 in the first round

of comments (his comment#2 in the first round). As mentioned by reviewer#1 above, it might be that certain other indicators (e.g. human development index or IFPRI's Global Hunger Index) could be relevant substitutes for certain indicators employed by us here but in choosing these particular indicators, we decided to stay consistent with what was proposed by previous peer-reviewed scientific publications.

In other words, we simply assumed that comments such as this one questioning the rationale for selection of indicators have already been tackled by previous studies. For those interested in further details on the rationale, we direct them to previous studies that we have relied upon (particularly Gustafson et al. 2016 and Acharya et al. 2014). In this manuscript, we mention these aspects first in the introduction section lines 62-65:

"These metrics were selected through consensus building activities involving a number of nutrition, economic, food system, and climate change experts representing a range of global public and private institutions (see Gustafson et al.¹⁸ and Acharya et al.¹⁹ for details).

And then in the 'limitation and uncertainties' sub-section lines 635-639:

"Our list of 25 indicators is not exhaustive, but a key criterion in the selection of these indicators is that they can be derived from data that are either directly available for all countries (e.g. food affordability & availability)³⁶ or can be readily estimated by processing global databases such as FBS²³ (e.g. Nutrient Balance Score - see methods above). This means that the indicator and metric scores can be calculated and regularly updated for subsequent years to track the progress of national food system sustainability."

Also, as mentioned by reviewer#1 above in his comment, this approach ensures - "the neatness of being consistent in building from those past studies." We also provide the Spearman correlation coefficient for all possible combinations of different indicators to give a quick idea on how they are related to each other (Supplementary Data S5).

Finally, we now added following lines (640-644) in the new subsection titled 'Limitations and uncertainties' within the method section to make this further explicit:

"We acknowledge that alternative indicators (e.g. human development index or IFPRI's Global Hunger Index) could be relevant substitutes for some employed by us here, but it was beyond the scope of this study to defend specific indicators or replace them, as this would have again entailed consensus building activities involving experts from diverse backgrounds. Instead, we simply utilized a complete set of indicators that had been fully described and justified in very recent studies^{18-20, 28} as being fully relevant for characterizing the status of global food sustainability."

Reviewer #4

Comment#3: The points raised by the reviewers seem to me to have been very well addressed. I just have some further minor comments:

l 12-13, I would suggest to replace "protein" by "food", as in some countries protein intake is well above needs.

l 107, 109, 153, 232, 257, 284, 363, 429, I suggest to replace "nation" by country; as the meaning of "nation" could differ depending on countries.

231, add "the" before "proportion"

252, add "the" before "resilience"

308, add "s" at the end of "pattern"

378, I would suggest to keep "food consumption" rather than "food systems" as what is dealt with here is only the impact from consumption. There are other means of reducing impacts on the environment in the food system, acting on transport, conservation, transformation, agricultural production...

399-400, I would strongly suggest to delete "except in the special case of subsistence farming". Why wouldn't their economic health be relevant? So called subsistence farming is not immune to economy and, most of the time, also linked to it.

436 and 444, suggest to replace "patterns" by "results"

441 Either delete "the" before "scaling up" or add "of" after it.

453 suggest to add "or not" after "inclusion"

Response: We thank the reviewer for these suggestions and have now made these changes.

Reviewer #5

Comment#4: Thank you for your responses to my comments. In all but one instance your responses have been satisfactory and the manuscript has been improved. However, I still have a concern regarding the way that bluewater use is estimated for the paper. The authors have included the following in the revised manuscript: "For calculating the national blue water footprint, we follow the bottom-up approach. We first import the global average blue water footprints (liters g⁻¹) of 94 FAO food items available from Mekonnen & Hoekstra and multiply them with their respective intake amounts (in g capita⁻¹ day⁻¹) in each country for the year 2011. Summing these up provided the diet related national average daily blue water footprint (in liters capita⁻¹ day⁻¹)."

I think the term 'bottom-up' is misleading here. The authors use it as the country-level estimate of bluewater use is the sum of the bluewater use for each of the food commodities consumed in the country. However, it implies that the global estimate of bluewater use is a sum of region-specific bluewater use, and this is not the case (as global estimates of the bluewater use are used at the local level). So I would remove the term 'bottom-up approach'.

Response: We agree with the reviewer and have now removed the term 'bottom-up approach' from the sentence in the method section line 565.

Comment#5: Secondly, the authors suggest that they used global estimates of bluewater use as these were the only estimates that were available and justify this by saying that this is common practice in the scientific literature. However, I am aware of some projects that use region-specific bluewater use estimates (e.g. the IMPACT model developed by IFPRI has region-specific estimates in its water model). This is a significant limitation of the study and it should be mentioned in the discussion section.

Response: We now acknowledge this limitation by adding following sentence at lines 656-658 in the 'limitations and uncertainties' sub-section:

“On the environmental side, one limitation of this study is that we used global average blue water footprint values for different food items.^{26,27} Using country-specific footprint values for individual food items (where available) will improve the accuracy of the results.”